# Exploring motivated reasoning in polarization over the unfolding 2023 judicial reform in Israel
Dora Simunovic [1,2,3] ✉, Anna Dorfman[2,4,5] & Maayan Katzir [3,4,5]

This work explored polarization over Israel's Judicial Reform, introduced in January 2023. We find that the reform divided people into pro- and anti-reform camps, which differed in characteristics such as institutional trust, patriotism, and national identity. For example, the camps disagreed about trust in the government versus the judiciary. In line with motivated reasoning—biased reasoning processes used to reach desired conclusions—people's pre-existing characteristics motivated polarized views of the reform as a threat to democracy (issue-based polarization) and negative emotions towards opponents (affective polarization). Further demonstrating a motivated process, pro-reform participants (the electorate majority), prioritized majority rule over other democratic features (e.g., minority rights) compared to anti-reform participants. Polarization differentially predicted downstream consequences (e.g., protest methods), indicating that the camps' reactions were motivated by the extremity of their views and negative emotions. This work extends the understanding of potentially motivated polarization processes and their immediate downstream consequences.

Polarization is a global challenge which has set back the so-called democratic wave of the 1980s and 1990s, locking it into stagnation and backsliding[1,2]. Mass polarization (polarization hereafter), occurring between the voting population rather than political elites[3], has been related to a variety of negative consequences, including decreased political participation[4], lower general trust[5], and intergroup conflict[6,7]. The 2023 judicial reform in Israel is an especially concerning example of such processes, as it involves polarization over an issue that reflects the way people understand democracy at its core. In this case, people do not merely disagree about a specific issue, like abortion or migration. Rather, they disagree on how such issues ought to be resolved in a democracy. Specifically, whereas some emphasize separation of power and a system of checks and balances between branches of government as central to democracy, others may regard majority rule as the key democratic principle. In this work we suggest that these and other polarized perceptions and opinions constitute motivated reasoning—reasoning processes used to reach desired conclusions and maintain desired views and opinions[8]. We show that polarization may stem from motivated opinions (e.g., politicized institutional trust) and is associated with downstream consequences such as endorsement of extreme protest methods and more pessimistic views of conflict resolution.

In January 2023, Israel's newly elected government, led by Prime Minister Binyamin Netanyahu, made a series of propositions (hereafter, the judicial reform) which would diminish the role of the Supreme Court and curtail the power it can exercise over the country's executive branch[9,10]. Although some groups in Israel expressed support for the proposed reform, to many local and international observers and stakeholders, the reform was a clear example of ongoing democratic backsliding, if not a soft coup d'état[11,12]. Of note, the parties comprising the government received only 48.38% of valid votes[13], indicating that although the government (i.e., the executive) held the majority in the parliament (i.e., the legislative) and therefore could enforce the reform, it did not enjoy widespread support among voters. In response to this crisis, protests erupted across the country, expressing concerns regarding broad and far-reaching negative impact of the reform, including healthcare services and academic freedom[14–16]. These protests forced the government to temporarily halt the reform and enter into dialogue with representatives of the opposition on March 27, 2023. At the same time, the government proceeded to promote the legislation unilaterally. The elite-directed dialogue broke down by June 14, 2023. On July 11, 2023, the government passed the first substantial part of the reform, eliminating "reasonableness" as a criterion the Supreme Court can use to curb government decisions[17].

[1]Bremen International Graduate School of Social Sciences, Constructor University, Campus Ring 1, Bremen, Germany. [2]Department of Psychology, Bar-Ilan University, Ramat Gan, Israel. [3]Conflict Management, Resolution & Negotiation Program, Bar-Ilan University, Ramat Gan, Israel. [4]These authors contributed equally: Anna Dorfman, Maayan Katzir. [5]These authors jointly supervised this work: Anna Dorfman, Maayan Katzir. ✉e-mail: doras@bigsss-bremen.de

The controversial judicial reform provides a unique opportunity to observe a newly emerging, dynamic, polarizing political issue as it unfolds. Understanding this specific issue is relevant globally as many regimes that self-identify as democracies have undergone similar processes (e.g., Poland, India, US, Brazil, Hungary, Turkey). Previous work suggested that leaders like Hungary's Viktor Orban have been actively using anti-liberal populist rhetoric[18] to undermine their country's institutions, particularly the electoral system, judiciary, and academia[19] contributing to the process of democratic backsliding and dismantling the cultural and institutional aspects of democracy[20]. As such processes are taking place around the world[21–23], it seems that no country is immune to the consequences of anti-liberal-democratic movements.

Even before the novel crisis centering around the judicial reform, Israel was a conflicted nation[24,25]. Conflicts can be traced to the ethno-religious disputes plaguing the country since its inception[26,27] and the challenge posed to democracy by religious institutions' involvement in secular governance[28]. Further, there is an ongoing controversy on how to conceptualize Israel's regime, as Israel's ethnic democracy model[29,30] has already been critiqued for enabling majority tyranny, unlike the liberal democracy model which features comprehensive protections for minority groups[31,32]. Importantly, tensions between the executive and judiciary branches have been building for years[33], peaking with the current far-right, conservative government. What sets this crisis and polarizing issue apart from previous conflictual issues is the general population's high level of engagement, as well as the stakes the reform poses for their country's continuing democracy. We argue that the newly proposed reform quickly polarized Israelis into opponents and supporters of the reform. Opponents presumably view the reform as a threat to democracy whereas supporters do not. We further suggest that this quick divide sparked by the reform was motivated by pre-existing beliefs (e.g., institutional trust) and identities (e.g., ethno-religious identity) shaping opponents' and supporters' conclusions and opinions about the reform.

When forming opinions and views about new information, people tend to engage in biased reasoning and information processing, motivated to align with pre-existing beliefs[8]. Numerous studies supported the notion that this tendency is not directed towards reaching accurate conclusions, but rather towards confirming and preserving desired views or ideological goals, for example positive views of one's political ingroup[34–38]. Motivated reasoning can exacerbate polarization, as it undermines processing of information that contradicts existing beliefs[39] and cements negative views of outgroups[40]. Biased information processing can enhance polarization over various policies by making supporters and opponents of an issue ignore information that does not align with their beliefs[41]. Negative views of outgroups can increase animosity between partisans by, for example, justifying harmful behavior against political opponents[38,41,42]. In this work we explore how pre-existing beliefs shape polarization over a new issue. For example, issue-based polarization over the judicial reform and affective polarization might be motivated by reform supporters' high trust and reform opponents' low trust in the government. We further explore how polarization motivates views on immediate downstream consequences. For example, endorsement of extreme protest methods by opponents with extreme views might be motivated by their side's continuous protests against the reform.

As the reform aims to shift the power balance between the executive government and the judiciary, trust in democratic institutions—judiciary, executive government, parliament, and media—is an obvious key variable to explore. It has been suggested that the relationship between trust in democratic institutions and polarization follows a negative feedback loop[43]. Empirical findings show that institutional trust relates to lower levels of polarization[44–46]. Dovetailing on the notion of motivated reasoning, institutional trust can itself be politicized and polarized[47–50], contributing to polarization in ideologically motivated ways. Although polarization can decrease institutional trust[51], in the case of the Israeli judiciary reform, politicized institutional (dis)trust has preceded polarization over the reform, as (dis)trust existed prior to the rolling out of the reform (January 2023), at least among certain groups[52,53]. Moreover, the judicial reform has been rolled out by the government to target the judiciary. As such, peoples' pre-existing

trust in the government and the judiciary likely preceded and drove their initial responses to this reform, and not vice versa. Importantly, although the relationship between institutional trust and polarization likely takes the shape of a negative feedback loop over time, in the case of polarization over the judicial reform it is more plausible that this negative loop started with pre-existing institutional (dis)trust than with polarization over the newly introduced reform. Therefore, we explored institutional trust as a predictor rather than an outcome of polarization over the reform.

Two other likely candidates to motivate people's views on the reform and polarize them are patriotism and national identity. We suggest that individuals who hold different views on patriotism and emphasize different national identities[54] are likely to be polarized in their views on the reform. Patriotism is often split into blind patriotism (sometimes equated with nationalism)[55,56], each representing a way of engaging with national identity and expressing loyalty for the associated community, institutions, and symbols. Blind patriotism is a rather extreme and unwavering expression of positive national affection. It is characterized by feelings of superiority and pride[57,58], as well as a suppression of ambivalent or critical attitudes toward the nation[59]. Blind patriotism is exclusive, limiting the acceptance of certain groups into the scope of the nation and its representative culture. It is associated with high levels of identification with the dominant ethnic group and is often fused with a religious identity[60,61]. In Israel, nationalities are significantly confounded with ethno-religious identities[61,62] (e.g., Jewish for the majority group). These identities are perceived as exclusive, genetically-based and immutable[63,64], especially by religious groups[65], present a relevant axis of political decision-making[66], and are different from the inclusive civic identity (i.e., Israeli) which is common to all citizens[61].

Different from blind patriotism, constructive patriotism reflects a more inclusive view of national belonging and manifests in acceptance of critical statements about one's nation, its history, culture, politics, and people. As such, it is sometimes lauded as the benevolent side of national attachment, in that it allows for maintaining positive views of one's own people without derogation of others[67]. It upholds a more civic, rather than ethno-religious, view of national belonging and is associated with liberal, universalist values[68].

Whereas the link between trust in institutions and polarization is well established, little is known about the role of patriotism in polarization. An exception is polarization over migration[69,70]. For example, one study found that appeals to ethnic identity and blind patriotism can polarize voters over immigration[69]. As right-wing populist narratives often use national superiority to justify anti-liberal-democratic policies[18], blind patriotism (i.e., unwavering national pride) and ethno-religious identity (i.e., Jewish) are likely to motivate biased support in an anti-democratic reform. On the other hand, constructive patriotism (i.e., accepting criticism of Israel) and the inclusive civic identity (i.e., Israeli) are likely to motivate biased opposition to it.

So far, we introduced factors that are likely to underlie motivated views on the reform and thus contribute to polarization. Additionally, we explored relevant psychological characteristics—generalized trust[71] and universalism-benevolence values[72]. Previous research found a mutually reinforcing negative relationship between generalized trust and polarization[73–75]. Given its documented role in polarization, we included generalized trust as a person-oriented parallel to institutional trust[76]. Universalism and benevolence are two closely related social values which denote other-regarding preferences, prosocial orientation, and a willingness to extend such behaviors across group lines[68]. Universalism and benevolence values have been related to more liberal, left-leaning policies across national contexts[77]. As the judicial reform undermines liberal democratic values, we explored the potential role of endorsing universalism/benevolence values in polarization over the reform.

To examine polarization, we utilized a group-based approach[78] and identified two camps—reflecting support of (pro-reform) and opposition to (anti-reform) the reform. We suggest that difference in views between these camps can motivate biased self-serving perceptions, both of others' position

on the reform as well as of the core issue at hand (i.e., understanding of democracy). First, self-serving motivated reasoning suggests that each camp is motivated to believe that most others hold a similar opinion, thus displaying the false consensus effect[79]. Second, even the importance each camp assigns to core features of democracy (e.g., majority rule, minority rights) is likely to be motivated to align with their polarized positions on the reform. For example, those whose current power position relies on majority rule principle are more likely endorse it than those who are currently harmed by this principle.

Finally, we examined potential downstream consequences of polarization that can be detrimental to social cohesion and impact intergroup relations. It is likely that people's views on the reform (i.e., pro- vs. anti-reform) motivate how they would perceive the legitimacy of various responses to it. Whereas much work has been conducted on the long-term consequences of polarization (e.g., lower political interest, decision-making gridlock, erosion of democratic processes, and lower general trust[4–6]), little is known about the more immediate, practical effects of polarization amid an ongoing social conflict (but see ref. 80). To fill this gap, we investigated four central groups of outcomes that are particularly relevant to the current context: endorsement of extreme protest methods (e.g., general strike), support for protest control (e.g., stun grenades), attitudes to conflict management strategies[81,82] (e.g., willingness to compromise), and delegitimization of political opponents (e.g., questioning their understanding of democracy). Our use of cross-sectional data precludes us from inferring causality; however, we argue that it is nevertheless useful to speak in terms of downstream consequences, as these consequences emerged after the judicial reform had split society into camps. Note, all the consequences we investigated may also act as reinforcers of polarization, thus creating a positive feedback loop.

In the current research, we adopted a data-driven exploratory approach to study polarization over the controversial judicial reform and its possible downstream consequences. Because we argue that the disagreements about the reform reflect contradictory views of democracy, our key dependent variable assessed the extent to which people view the reform as a threat to democracy. Our analysis showed that the reform split people into two opposing camps. We examined pre-existing differences between these pro- and anti-reform camps in national identity, patriotism, and institutional trust, as well as potentially motivated differences in perceptions between them (i.e., importance of democratic principles, false consensus). We extended the scope of our investigation to three types of polarization—issue-based[78] (i.e., extremity of one's view pro or against the reform), affective[6,83] (i.e., extent of negative emotions towards political opponents), and perceived societal polarization[84] (i.e., extent to which one views society as polarized in general.) Issue-based and affective polarization touch directly on the reform and the resulting pro- and anti-reform camps, whereas perceived societal polarization goes beyond the specific issue at hand. As little is known about how each of these polarization types relates to national identity, patriotism, and institutional trust (but see ref. 44 for affective polarization and institutional trust), we examined how the aforementioned pre-existing differences between camps relate to each type of polarization. Finally, whereas most previous research showed that all types of polarization are associated with negative long-term consequences (e.g., loss of social cohesion[85], negative partisanship[86], democratic backsliding[87,88]), we focused on shorter-term downstream consequences.

To summarize, we focused on the following research questions:

Q1: How do institutional trust and patriotism relate to individuals' views on the unfolding, polarizing issue of the judicial reform?

Q2: How do people from different camps differ (e.g., in degree of polarization, in prioritizing central features of democracy)?

Q3: How do institutional trust and patriotism predict different types of polarization (i.e., issue-based, affective, and perceived societal), for pro- and anti-reform camps?

Q4: How do different types of polarization predict immediate downstream consequences (e.g., conflict management strategies) amid an unfolding social conflict for pro- and anti-reform camps?

Our research questions stem from two broad pre-registered aims. Specifically, we aimed to understand Israeli Jews' polarized views on the reform (Q1 and Q2) and to examine potential downstream consequences of polarization (Q4). Another, more specific, pre-registered aim was to examine potential shifts in views on the reform. For this we collected data in two waves set approximately two months apart. Given that shift in views was relatively minor and only significant for the pro-reform camp, we report this finding in the Results section and do not discuss it. In addition to the pre-registered aims, as the literature addresses different types of polarization, we used the same predictors as those used to understand Israeli Jews' polarized views on the reform to understand what predicts issue-based, affective, and perceived societal polarization (Q3).

## Methods
### Sample, recruitment, and exclusions
We collected data in two waves. In the first wave (T1; March 9–12, 2023), we recruited 822 participants via the Midgam Project Web Panel, an Israeli polling company, to take part in a large study on intergroup relations. We sampled a representative sample (in terms of age and gender) of only the majority Israeli population, as we did not sample two hard-to-reach minority groups (Ultra-Orthodox Jews and Palestinian citizens of Israel). To understand the situation in more depth, we ran a pre-registered follow-up wave (T2; May 8–11, 2023, pre-registered on May 8, https://aspredicted.org/JMP_QKT), contacting only 709 participants who completed the T1 survey and passed a pre-registered attention check. The majority ($N = 584$) responded to the call. This loss of respondents should not be regarded in terms of a classic drop-out rate because at T1 participants were unaware that we will contact them again at T2.

We excluded 30 participants who failed one pre-registered criterion (completed the survey in less than 4 min). Of the remaining 554 participants, 56 participants had incomplete data and could not be analyzed (i.e., political affiliation, demographic data, dependent variables). The final sample included 498 Israeli Jews (249 women, 249 men; ages 18–64, $M_{age} = 42.27$, $SD_{age} = 12.10$). Information on gender was provided by participants (see Table 1). The majority identified as secular (61.04%). Regarding political orientation (1 = *extreme left*, 5 = *center*, 9 = *extreme right*), 7.23%, identified as left-leaning (1–3), 55.62% as center (4–6), and 37.15% as right-leaning (7–9).

In addition to the pre-registered exclusion we applied, we also pre-registered exclusions based on identical responses to opposing items. Upon additional review we realized that identical responses to these items do not necessarily indicate low-quality responses that merit exclusion (e.g., a person who disagrees with both the item *The reform is crucial for Israel's democracy* and the item *The reform is a threat to democracy*), as these items are not logically opposing. Moreover, excluding participants based on this criterion reduced the sample by 40.6%. Therefore, we decided to forgo this exclusion criterion. For transparency, we reproduced the core analyses using the original pre-registered criteria and present the key figures and tables in Supplementary Methods 5.

Participants were compensated separately for each survey, as per the panel's guidelines. This study was approved by the Psychology Department Ethic Review Board at Bar Ilan University (#2023/49/1 and #2023/30). We followed all relevant ethical regulations and obtained informed consent from participants.

### Procedure and materials
Table 1 presents study variables, including example items, number of items for each variable, Cronbach's α, scales, and the wave in which the variable was assessed (see Supplementary Methods 1 in the SI for a full list of translated items).

**Pre-existing factors**. As pre-registered, we assessed *institutional trust* (in judiciary, government, parliament, and media) in T1 and T2. *Blind* and *constructive patriotism*[89], *civic identity* (i.e., identification with Israelis), and *ethno-religious identity* (i.e., identification with Jews) were

**Table 1 | List of constructs used in the study**

| Construct | # Items | Sample item | Scale (min and max) | Wave |
|---|---|---|---|---|
| Generalized trust[A] | 3 α = 0.69 | Most people can be trusted. | 1 = You can't be too careful 7 = Most people can be trusted | T2 |
| Universalism/ Benevolence[B] | 5 α = 0.81 | It's very important to him (her) to help the people around him (her). He (she) wants to care for their well-being. | 1 = Not like me at all 6 = Very much like me | T1 |
| Blind patriotism[C] | 5 α = 0.77 | People should support their country even when the country is in the wrong. | 1 = Disagree completely 5 = Agree completely | T1 |
| Constructive patriotism[C] | 3 α = 0.74 | The world would be a better place if Israelis acknowledged Israel's shortcomings. | 1 = Disagree completely 5 = Agree completely | T1 |
| Civic identity[C] | 1 | How emotionally close do you feel to Israelis? | 1 = Not at all 7 = To a very high extent | T1 |
| Ethno-religious identity[C] | 1 | How emotionally close do you feel to Jews? | 1 = Not at all 7 = To a very high extent | T1 |
| Institutional trust[D] | 4 | To what extent do you trust the following social institutions: Government | 1 = Distrust completely 5 = Trust completely | T1,T2 |
| Reform-as-threat views[E] | 1 | The reform is a threat to democracy. | 1 = Disagree completely 7 = Agree completely | T1,T2 |
| Issue-based polarization[E] | 1 | Absolute deviation of response to the reform-as-threat item from the 7-point scale's mid-point (4). | 0 = neutral view 3 = extreme view | T1,T2 |
| Affective polarization[E,F] | 8 α = 0.88 | To what extent do you experience the following emotions when thinking about supporters/opponents of the reform: Hate | 1 = Not at all 7 = To a high extent | T2 |
| Perceived societal polarization[D] | 1 | To what extent do you think Israeli society is polarized? | 1 = Not at all polarized 7 = Extremely polarized | T2 |
| False consensus[E] | | | | |
| % pro-reform | 1 | What percentage of Israelis do you think support the reform? | | T2 |
| % anti-reform | 1 | What percentage of Israelis do you think oppose the reform? | Total percent must add to 100 | T2 |
| % no opinion | 1 | What percentage of Israelis do you think have no opinion on the issue of the reform? | | T2 |
| Features of democracy[E] | 6 | Majority rule. | 1 = Not at all important 7 = Important to a high extent | T2 |
| Conflict management strategies[E] | 7 α = 0.84 | I am ready to give up some of my principles so that we do not end up in a civil war over the issue of the reform. | 1 = Disagree completely 7 = Agree completely | T2 |
| Protest methods[E] | 3 α = 0.87 | Civil revolt | 1 = Not legitimate at all 7 = Very legitimate | T2 |
| Protest control[E] | 3 α = 0.81 | Arrests | 1 = Not legitimate at all 7 = Very legitimate | |
| Delegitimization[E] | 6 α = 0.83 | They do not understand what democracy is. | 1 = Not characteristic 7 = Highly characteristic | T2 |
| Religiosity[E] | 1 | How religious are you? | 1 = Not religious 2 = Somewhat religious 3 = Religious | T1 |
| Political orientation[E] | 1 | What is your political orientation? | 1 = Far left 5 = Center 9 = Far right | T1,T2 |
| Gender[E,a] | 1 | What are your preferred pronouns? | 1 = feminine form 2 = masculine form | T1 |
| Birth year[G,b] | 1 | What year were you born? | | |
| Education[G] | 1 | What is your highest level of education? | 1 = High school 2 = Post high school diploma 3 = B.A. 4 = M.A. 5 = Ph.D. | |
| SES[G] | 1 | Please describe your monthly income. | 0 = No income 5 = Significantly above average | |

*Note.* The full list of translated items is available in Table S1 in the SI.

[a]For gender inclusivity, we had two versions of the survey—one addressed women, the other addressed men. This is because Hebrew is a gendered language. The default is masculine and to avoid possible biases, we wanted to address women as well.

[b]To calculate participants' age, birth year was subtracted from 2023 (year of study).

Sources: A = ESS Round 8: European Social Survey Round 8 Data (2016); B = Schwartz (2007); C = International Social Survey Programme: National Identity III - ISSP 2013; D = Varieties of Democracy (V-Dem) Project; E = item created for the purpose of this investigation. F = Adapted from: Iyengar et al. (2019). G = Midgam Panel item.

only assessed in T1. Additionally we assessed relevant psychological characteristics: adherence to *universalism/benevolence values*[90] (T1) and *generalized trust* (T2). We received demographic information from the panel, except for *political orientation*, gender, and religiosity.

**View of reform as threat**. We assessed participants' views of the reform as a threat to democracy in T1 and T2. Inspecting the distribution of responses to this item provides a method for observing polarization and therefore we pre-registered views of the reform as a threat to democracy as one of the key dependent variables.

**Issue-based polarization**. Issue-based polarization manifests when people hold extreme positions regarding the reform-as-threat item. Specifically, if people are polarized about the issue, many people will perceive the reform as no threat at all (e.g., select "1" in response to this item, on the 7-point scale) and many others will perceive the reform as a great threat (e.g., select "7"). To assess issue-based polarization more directly, we created a measure of how extreme participants' views were by calculating the absolute deviation of each participant's response to the reform-as-threat item (in T1 and in T2) from the 7-point scale's mid-point (4). We used the mid-point for two reasons: (1) to avoid underestimating the possibility one cluster tends more towards the extreme ends of the scale than the other, and (2) to ease interpretation of views extremity. Issue-based polarization reflects the degree to which views of the reform-as-threat are polarized (i.e., tend towards the extremes and away from the neutral position).

**Affective polarization**. Participants rated their negative (5 items) and positive (3 items, reverse-coded) emotions both towards those who support and those who oppose the reform, as to not bias participants' responses. For each participant, we used only the items reflecting emotions towards the opposing camp. Higher scores indicated more negative affect towards the opposing camp.

**Perceived societal polarization**. Participants rated the extent to which they think Israel is polarized, without mentioning the reform. Higher scores indicated higher perceived polarization.

**Motivated perceptions**. We assessed two perceptions that can be motivated by participants' views on the reform—false consensus and importance of democratic features. To assess *false consensus*, we looked at participants' perceptions of the distribution of opinions about the reform in the population. Participants indicated the percentage of the population that supports, opposes, and has no opinion about the reform (adding up to 100). False consensus manifests when participants estimate their own camp (either pro- or anti-reform) as larger than the other camps. To assess *importance of democratic features*, we presented participants with six features—majority rule, separation of power between branches of government, protection of minority rights, independent and free media, free elections, and protection of human rights for everyone equally. Participants rated the extent to which they believe each feature is important to Israel's democracy.

**Motivated downstream consequences**. Participants rated their opinions regarding downstream consequences that can be motivated by the extremity of their views on the reform (i.e., issue-based polarization) and their feeling towards opposing camps (i.e., affective polarization). Following Exploratory Factor Analysis (see Supplementary Methods 4 in SI for full description), we focused on four immediate downstream consequences of polarization: endorsement of extreme *protest methods* (e.g., civil revolt), support of extreme *protest control* (e.g., police violence), *conflict management strategies* (e.g., compromise, dialogue), and *delegitimization of political opponents*. *Delegitimization* was assessed in the context of the judicial reform. Participants rated characteristics of people who support the reform and of people who oppose it (e.g., "They do not

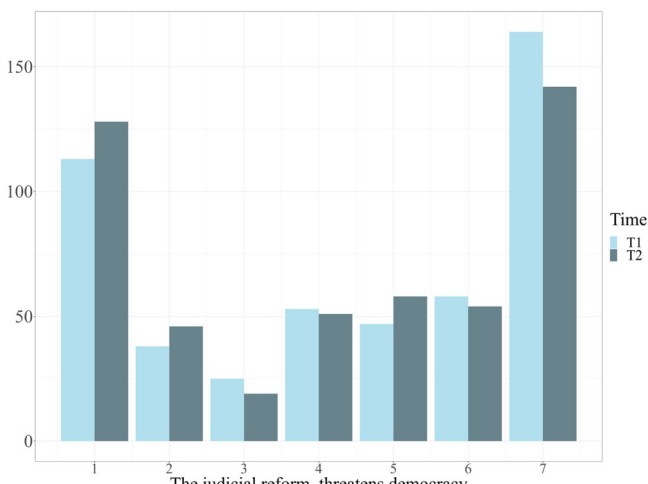

**Fig. 1 | Distributions of responses to the statement "Reform is a threat to democracy" at T1 and T2.** Time 1 (T1) distribution appears in light blue, Time 2 (T2) distribution appears in dark blue. Responses ranged from 1 = *Disagree completely* to 7 = *Agree completely*. *N* = 498.

understand what democracy is", "They want revenge"). As with affective polarization, for each participant, we used only the items reflecting delegitimization of the opposing camp. Higher scores indicated greater delegitimization.

## Statistics and reproducibility
We pre-registered hierarchal (multi-level) linear regressions as the analytical tool to answer our main questions. We conducted additional analyses (e.g., cluster analysis, Welch's t-tests, mixed ANOVAs) when a more comprehensive exploration was required. All tests were two-tailed. All data was analyzed using R software 4.3.0 in RStudio 2023.03.1. Distributions of errors were assumed to be normal, but this was not formally tested. We applied the Greenhouse-Geiser correction for the sphericity assumption.

## Results
We present the results in four sections, corresponding to the four research questions. To answer Q1, we established the polarized nature of views on the reform and its association with pre-existing factors (e.g., institutional trust, generalized trust). We also examined potential shifts in views across data collection waves. To answer Q2, we used cluster analysis to group participants based on their views of the reform and political orientation. We then compared the two clusters that emerged (pro- vs. anti-reform) on pre-existing factors, motivated perceptions, and downstream consequences. To answer Q3, we examined whether pre-existing factors predict issue-based, affective, and perceived polarization, and whether these associations vary between the clusters. To answer Q4, we examined how different types of polarization motivate various downstream consequences (e.g., endorsement of extreme protest actions).

### Q1: Understanding views on the reform
**Observing polarization**. First, to establish that the judicial reform is indeed a highly polarized issue, we examined the distribution of participants' agreement with the reform-as-threat item. The distribution deviated from normality (T1: Shapiro–Wilk test: $W = 0.83$, $p < 0.001$; kurtosis: $-1.50$, $SE = 0.22$; T2: Shapiro–Wilk test: $W = 0.83$, $p < 0.001$; kurtosis: $-1.58$, $SE = 0.22$), and is best described as bimodal, thus reflecting polarization (Fig. 1). Although the reform was viewed, on average, as more threatening in T1 ($M = 4.43$, $SD = 2.40$) than in T2 ($M = 4.19$, $SD = 2.41$; $t(497) = 3.61$, $p < 0.001$, $d = 0.10$), this difference was driven by a minority of participants. Most participants (56.22%) reported identical views across T1 and T2. For 33.53% of the sample,

**Table 2 | Predicting views of the reform as threat**

| Variable | B | 95%CI | t | p |
|---|---|---|---|---|
| Intercept | 4.32 | [4.21, 4.43] | 77.37 | <0.001 |
| Gender | **0.12** | **[0.01, 0.24]** | **2.15** | **0.032** |
| Age | −0.01 | [−0.13, 0.11] | −0.10 | 0.917 |
| Education | **−0.12** | **[−0.23, 0.00]** | **−1.96** | **0.050** |
| SES | 0.08 | [−0.04, 0.20] | 1.31 | 0.191 |
| Religiosity | **−0.38** | **[−0.50, −0.25]** | **−5.70** | **<0.001** |
| Wave (T1 vs. T2) | **−0.18** | **[−0.25, −0.11]** | **−5.26** | **<0.001** |
| Universalism/Benevolence | −0.02 | [−0.13, 0.10] | −0.27 | 0.788 |
| Generalized trust | −0.05 | [−0.17, 0.06] | −0.90 | 0.366 |
| Political affiliation | **−0.17** | **[−0.30, −0.05]** | **−2.79** | **0.005** |
| Civic identity | **0.25** | **[0.10, 0.40]** | **3.27** | **0.001** |
| Ethno-religious identity | −0.14 | [−0.30, 0.02] | −1.72 | 0.086 |
| Constructive patriotism | **0.45** | **[0.32, 0.59]** | **6.59** | **<0.001** |
| Blind patriotism | **−0.20** | **[−0.35, −0.04]** | **−2.50** | **0.013** |
| Trust in judiciary | **0.57** | **[0.45, 0.70]** | **8.95** | **<0.001** |
| Trust in government | **−0.62** | **[−0.75, −0.49]** | **−9.13** | **<0.001** |
| Trust in parliament | 0.01 | [0.30, 0.53] | 0.12 | 0.903 |
| Trust in media | **0.42** | **[−0.11, 0.13]** | **6.99** | **<0.001** |
| *Interaction with Wave (T1 v. T2)* | | | | |
| *Trust in judiciary | 0.06 | [−0.04, 0.15] | 1.18 | 0.238 |
| *Trust in government | −0.02 | [−0.12, 0.08] | −0.46 | 0.646 |
| *Trust in parliament | 0.01 | [−0.09, 0.11] | 0.13 | 0.898 |
| *Trust in media | 0.02 | [−0.07, 0.12] | 0.46 | 0.644 |
| AIC | 3507.03 | | | |
| BIC | 3624.33 | | | |
| Adjusted R² | 0.63 | | | |
| Sigma | 1.05 | | | |

*Note*. Stepwise regression is included in Supplementary Methods 3 in the SI. Significant effects are bolded.

responses deviated up to two scale points, either above or below their initial ratings, whereas only 10.25% changed their minds more drastically.

**Predicting view of the reform as threat**. To better understand views of the reform as a threat at T1 and T2, we fit a linear mixed model (LMM) predicting views from pre-existing factors—patriotism (constructive, blind), national identity (civic, ethno-religious), trust in democratic institutions (judiciary, government, parliament, media), and wave (T1 vs. T2). We controlled for demographic indicators (gender, age, education, subjective socio-economic status, degree of religiosity), political orientation, psychological characteristics (universalism/benevolence and generalized trust), and participants as a random intercept. To understand whether the relationship between institutional trust and views has shifted, we included interactions between wave and trust in each democratic institution. All predictors were mean-centered. Of note, this model examines predictors of views and does not yet address issue-based polarization.

The model (Table 2) explained a considerable proportion of the variance ($R_{Adj}^2 = 0.63$). Reflecting the heart of the current issue, greater trust in the judiciary was associated with higher perceived threat from the reform, whereas greater trust in government was associated with lower perceived threat. Greater trust in media was also associated with higher perceived threat. The more participants self-identified as left-leaning politically and reported greater levels of constructive patriotism and civic identity, the more

they viewed the reform as a threat. Lower levels of blind patriotism were associated with higher perceived threat. The effect of wave was significant, indicating a decrease in perceived threat from T1 to T2. However, the wave × institutional trust interactions were not significant.

**Q2: Characterizing pro- and anti-reform camps**

**Cluster analysis.** To cluster participants into camps reflecting their views on the reform we conducted a hierarchical clustering using the *factoextra* package in R. Given that the reform is a highly politicized issue, we clustered participants based on their political orientation and their views of the reform as threat (i.e., the polarizing issue) averaged across T1 and T2. Two clusters emerged as the optimal number of clusters, supporting the existence of two main camps. A visual inspection of the associated dendogram and the Elbow method (see Supplementary Methods 2, Figs. S1 and S2 in SI) supported the two-cluster solution. The anti-reform cluster ($N = 262$) comprised participants who view the reform as a threat to democracy ($M = 6.14$, $SD = 1.03$), and whose political orientations average to the center ($M = 4.99$, $SD = 1.22$). The pro-reform cluster was similar in size ($N = 236$), did not view the reform as a threat ($M = 2.29$, $SD = 1.44$) and was politically right-leaning ($M = 6.91$, $SD = 1.07$).

**Shifts in views by cluster.** To further explore the observed decrease in perception of threat found in the "predicting view of the reform as threat" analysis, we conducted a repeated-measured ANOVA with a 2 (cluster: pro, anti) x 2 (time: T1, T2) on views of the reform. We found a main effect of cluster $F(1, 496) = 1199.81$, $p < 0.001$, $\eta_p^2 = 0.71$, and a main effect of time, $F(1, 496) = 13.99$, $p < 0.001$, $\eta_p^2 = 0.03$, indicating that the anti-reform cluster viewed the reform as more threatening and that the overall perceived threat decreased from T1 to T2. These main effects were qualified by a cluster × time interaction, $F(1, 496) = 5.21$, $p = 0.023$, $\eta_p^2 = 0.01$. Pairwise comparisons indicated the shift was driven by pro-reform participants ($M_{T1} = 2.48$, $SD = 1.84$; $M_{T2} = 2.09$, $SD = 1.48$), $t = 4.15$, $p < 0.001$, $\eta_p^2 = 0.034$. There was no significant shift among anti-reform participants ($M_{T1} = 6.19$, $SD = 1.17$; $M_{T2} = 6.09$, $SD = 1.22$), $t = 1.06$, $p = 0.290$, $\eta_p^2 = 0.002$.

**Comparisons between clusters.** We compared clusters' pre-existing factors (i.e., psychological characteristics, identity, patriotism, and institutional trust), polarization, and motivated downstream consequences (Fig. 2). We found difference in pre-existing factors between the camps that might be differentially motivating. Pro-reform participants held stronger ethno-religious and civic identities than anti-reform participants. Notably, pro-reform participants held an especially strong ethno-religious identity (6.33 on a 7-point scale). Anti-reform (vs. pro-reform) participants expressed more constructive patriotism and less blind patriotism. They also indicated more trust in the judiciary and media, and less trust in the government and parliament than pro-reform participants. Extremity of one's view on the reform (i.e., issue-based polarization) was greater among anti-reform participants and they also perceived Israel as more polarized in general, as compared to pro-reform participants. There were also difference in downstream consequences; anti-reform participants simultaneously expressed greater support for conflict management and greater endorsement of extreme protest methods, but they supported extreme protest control methods less than pro-reform participants.

Clusters also differed in political orientation (Fig. 3). Pro-reform participants leaned politically further to the right than anti-reform participants, $t(495.61) = −18.71$, $p < 0.001$, 95%CI[−2.12, −1.71], $d = 1.67$. Yet, anti-reform participants identified across the entire left-right political spectrum, indicating that this issue goes beyond party loyalties and represents a different axis of disagreement.

**Motivated perceptions—false consensus.** We were interested in whether participants estimate their own camp (pro- vs. anti-reform) as

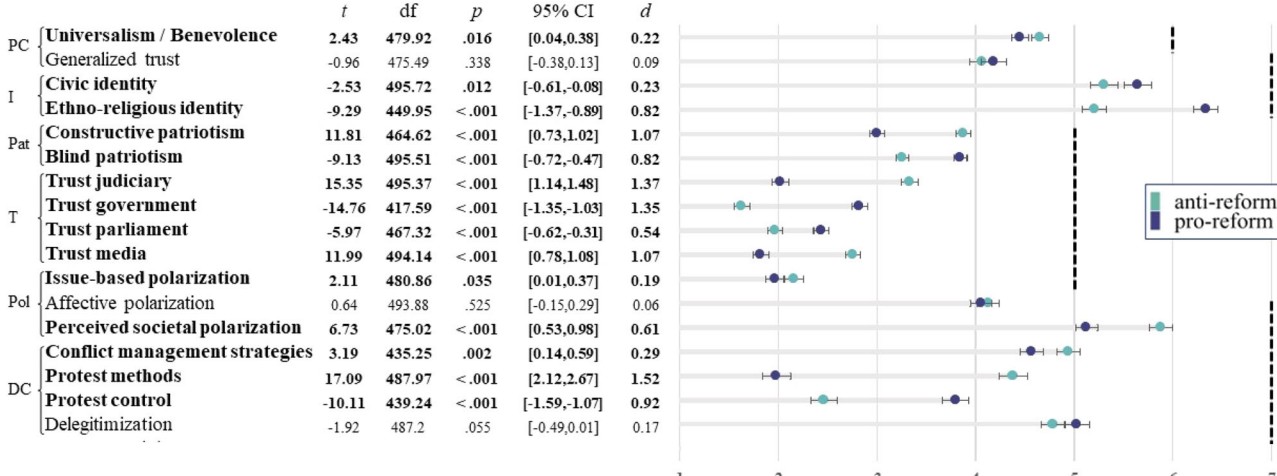

| | | t | df | p | 95% CI | d |
|---|---|---|---|---|---|---|
| PC | Universalism / Benevolence | 2.43 | 479.92 | .016 | [0.04,0.38] | 0.22 |
| | Generalized trust | -0.96 | 475.49 | .338 | [-0.38,0.13] | 0.09 |
| I | Civic identity | -2.53 | 495.72 | .012 | [-0.61,-0.08] | 0.23 |
| | Ethno-religious identity | -9.29 | 449.95 | <.001 | [-1.37,-0.89] | 0.82 |
| Pat | Constructive patriotism | 11.81 | 464.62 | <.001 | [0.73,1.02] | 1.07 |
| | Blind patriotism | -9.13 | 495.51 | <.001 | [-0.72,-0.47] | 0.82 |
| T | Trust judiciary | 15.35 | 495.37 | <.001 | [1.14,1.48] | 1.37 |
| | Trust government | -14.76 | 417.59 | <.001 | [-1.35,-1.03] | 1.35 |
| | Trust parliament | -5.97 | 467.32 | <.001 | [-0.62,-0.31] | 0.54 |
| | Trust media | 11.99 | 494.14 | <.001 | [0.78,1.08] | 1.07 |
| Pol | Issue-based polarization | 2.11 | 480.86 | .035 | [0.01,0.37] | 0.19 |
| | Affective polarization | 0.64 | 493.88 | .525 | [-0.15,0.29] | 0.06 |
| | Perceived societal polarization | 6.73 | 475.02 | <.001 | [0.53,0.98] | 0.61 |
| DC | Conflict management strategies | 3.19 | 435.25 | .002 | [0.14,0.59] | 0.29 |
| | Protest methods | 17.09 | 487.97 | <.001 | [2.12,2.67] | 1.52 |
| | Protest control | -10.11 | 439.24 | <.001 | [-1.59,-1.07] | 0.92 |
| | Delegitimization | -1.92 | 487.2 | .055 | [-0.49,0.01] | 0.17 |

**Fig. 2 | Comparison of pro- and anti-reform clusters on main study variables.** Results of t-tests comparing pro-reform (purple) and anti-reform (green) clusters on various study measures. Items were rated from 1 = *completely disagree* to 7 = *completely agree* (except for universalism/benevolence: 1–6; patriotism and institutional trust: 1–5). Dashed lines indicate upper scale limit. Variables with significant differences between clusters are bolded. PC = psychological characteristics; I = identity; Pat = patriotism; T = institutional trust; Pol = polarization; DC = downstream consequences. Circles denote cluster means and error bars denote 95% CIs. $N_{generalized\ trust}$ = 490, all other Ns = 498.

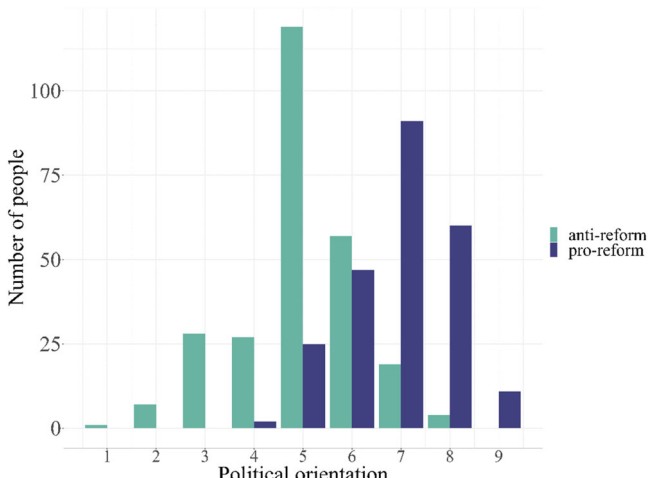

**Fig. 3 | Distribution of political orientation by cluster.** N = 498 participants indicated their political orientation on a 9-point scale (1 = *extreme left*, 5 = *center*, 9 = *extreme right*). Pro-reform cluster appears in purple, anti-reform cluster appears in green.

larger than the other camps (reflecting false consensus) and whether this biased perception might also be motivated by issue-based polarization (i.e., the extremity of one's view on the issue). To do that, we ran a mixed-model ANOVA on estimated camp size, with cluster (pro, anti) and issue-based polarization (0 = *neutral view*, 1 = *moderate view*, 2 = *extreme view*, 3 = *most extreme view*) as the between-participants factors and estimated camp (pro, no opinion, anti) at the repeated-measures factor.

First, the cluster × estimated camp interaction, $F(1.98, 971.45) = 53.24$, $p < 0.001$, $\eta_p^2 = 0.10$, indicated a false consensus effect. Specifically, the pro-reform cluster estimated that most people support the reform ($M = 50.82\%$, $SD = 18.84$) and that about one-third oppose it ($M = 32.34\%$, $SD = 13.25$). The anti-reform cluster estimated that most people oppose the reform ($M = 53.52\%$, $SD = 16.64$) and that about one-third support it ($M = 31.12\%$, $SD = 13.25$). Both clusters estimated the neutral opinion as uncommon ($M_{AntiReform} = 15.36\%$, $SD = 15.13$; $M_{ProReform} = 16.84\%$, $SD = 14.75$). This interaction was qualified by the cluster × issue-based polarization × estimated camp interaction, $F(5.95, 971.45) = 10.75$, $p < 0.001$, $\eta_p^2 = 0.06$ (Fig. 4).

People who held extreme views expressed greater false consensus than people who were less extreme. Follow-up interaction contrasts indicated that the cluster × estimated camp size interaction, which reflects false consensus, was significant for the most extreme participants, $F(2, 490) = 138.97$, $p < 0.001$, and the extreme participants, $F(2490) = 29.21$, $p < 0.001$. This two-way interaction was not significant for the moderate, $F(2, 490) = 0.75$, $p = 0.475$, and neutral participants, $F(2490) = 1.93$, $p = 0.146$.

Note that because the percentages that participants assigned to the three camps sum up to 100, there was no variance between participants across the different estimated camp repeated-measures factor. Therefore, the main effects of the between-participants variables and their interaction are uninterpretable (for similar analysis, see ref. 91). The interaction between issue-based polarization and estimated camp size is also meaningless, as it provides information about a person's extremity without the direction (pro- or anti-reform) of said extremity. Therefore, we reported above only the meaningful two-way and three-way interactions.

**Motivated perceptions—features of democracy.** We compared the clusters in terms of the importance they assigned to different democratic features (Fig. 5). Specifically, anti-reform (vs. pro-reform) participants assigned greater importance to five of the six features—separation of power between branches of government, $t(458.31) = 4.76$, $p < 0.001$, 95%CI[0.37, 0.88], $d = 0.43$, $N = 494$; protection of minority rights, $t(422.07) = 7.25$, $p < 0.001$, 95%CI[0.63, 1.11], $d = 0.66$, $N = 496$; independent and free media, $t(398.59) = 6.20$, $p < 0.001$, 95%CI[0.50, 0.97], $d = 0.57$, $N = 495$; free elections, $t(444.38) = 3.26$, $p = 0.001$, 95%CI[0.13, 0.54], $d = 0.30$, $N = 496$; protection of equal human rights, $t(396.78) = 7.43$, $p < 0.001$, 95%CI[0.68, 1.17], $d = 0.68$, $N = 495$. Majority rule was the only exception to this pattern—pro-reform participates (who are represented by the ruling majority—the government that proposed the reform) rated this democratic feature as more important than anti-reform participants, $t(493) = -4.22$, $p < 0.001$, 95% CI[−0.80, −0.29], $d = 0.38$, $N = 495$.

## Q3: Predicting different types of polarization
For each of the three polarization types, we fitted a linear model using the same predictors (mean centered) as those predicting views of the reform as threat (Q1): patriotism (constructive, blind), national identity (civic, ethno-religious), trust in democratic institutions (judiciary, government, parliament, media). In this analysis, we included cluster and excluded political orientation, as cluster took into account political orientation. We again

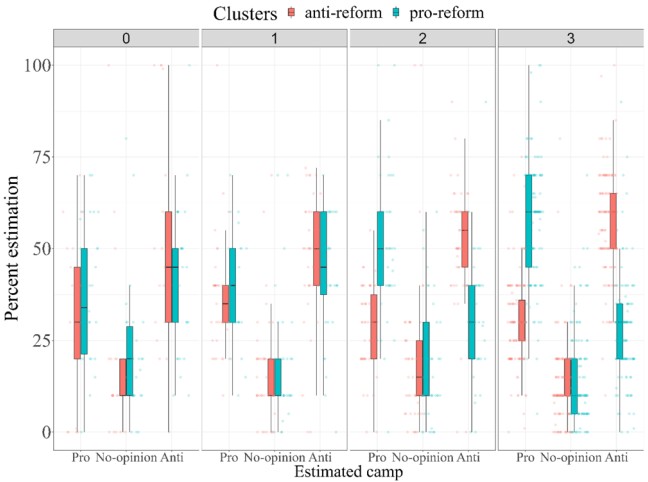

**Fig. 4 | False consensus by cluster and issue-based polarization.** Percent estimation of camp as a function of estimated camp (Pro = pro-reform, No-opinion, Anti = anti-reform), cluster (red: pro-reform, green: anti-reform) and level of issue-based polarization (0 = *neutral view*, 1 = *moderate view*, 2 = *extreme view*, 3 = *most extreme view*). $N$ = 498. Error bars denote 95% CIs.

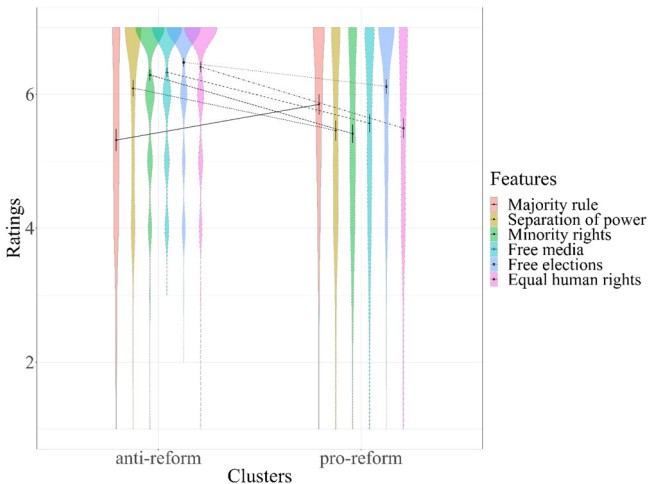

**Fig. 5 | Importance of democracy features.** Ratings of importance of democracy features (1 = *not important at all*, 7 = *very important*) by cluster. Majority rule (light orange), separation of power (mustard), minority rights (green), free media (turquoise), free elections (purple), and equal human rights (pink). $N_{Majority\ rule}$ = 495, $N_{Separation\ of\ power}$ = 494, $N_{Minority\ rights}$ = 496, $N_{Free\ media}$ = 495, $N_{Free\ elections}$ = 496, $N_{Equal\ human\ rights}$ = 495.

controlled for demographic indicators (gender, age, education, subjective socio-economic status, degree of religiosity) and psychological characteristics (universalism/benevolence, generalized trust). To further examine differences between camps, we included interactions with cluster. This allowed us to explore the motivated associations between each predictor and polarization. For example, for the pro-reform cluster, trust in government should positively associate with issue-based polarization (i.e., view extremity), reflecting that the more they trust the government, the more extreme their view (i.e., that the reform is not a threat) is; conversely, for the anti-reform cluster, trust in government should negatively associate with issue-based polarization, reflecting that the more they trust the government, the less extreme their view (that the reform is a threat) is.

**Issue-based polarization**. Issue-based polarization reflects how extreme participants' views of the reform are. Of note, whereas Q1 addressed views on the reform, the current analysis predicts the degree to which those views

are polarized. Because issue-based polarization was uniquely measured in both waves of data collection, we fit a linear mixed model (LMM), adding wave as a fixed effect and participants as a random intercept.

As Table 3 (Model 1) shows, in the anti-reform cluster, civic identity and constructive patriotism predicted more polarization. In the pro-reform cluster, ethno-religious identity predicted more polarization and constructive patriotism predicted less polarization. Institutional trust might also reflect that polarization is motivated. Specifically, in the anti-reform cluster, more trust in government predicted less polarization whereas more trust in media predicted more polarization. In the pro-reform cluster, more trust in government predicted more polarization whereas more trust in the judiciary predicted less polarization.

**Affective polarization**. As Table 3 (model 2) shows, across clusters, universalism/benevolence predicted less affective polarization. More constructive patriotism predicted more affective polarization in the anti-reform cluster. Trust in the judiciary, government, and media showed the same pattern as in issue-based polarization, suggesting that affective polarization might be motivated by institutional trust similarly to issue-based polarization.

**Perceived societal polarization**. In line with the notion that perceived societal polarization does not directly concern the current issue, we found that more universalism/benevolence values and less generalized trust associated with more polarization across clusters. Unlike issue-based and affective polarization, for perceived societal polarization the associations with national identity, patriotism, institutional trust, as well as their interactions with cluster were non-significant.

**Q4: Predicting downstream consequences of polarization**
Finally, we examined how issue-based, affective, and perceived societal polarization are associated with immediate downstream consequences. We included interactions of polarization types with the cluster variable to examine whether a given motivated consequence was differently associated with polarization for each cluster (i.e., suggesting that polarization could have driven participants from different clusters in opposing directions, reflected in a significant interaction term), or the association was similar for both clusters (reflected in main effects of polarization). From the perspective of motivated reasoning, we were mainly interested in issue-based and affective polarization which directly relate to the issue at hand.

**Downstream consequences**. We fitted linear models controlling for demographic characteristics as in previous analyses. Following an Exploratory Factor Analysis (see Supplementary Methods 4 in SI), we focused on four immediate downstream consequences of polarization. The results are presented in Table 4. As we already reported effects of cluster on downstream consequences when answering Q2 above, we do not reiterate them here.

Protest methods. Perceived polarization was associated with greater endorsement of extreme protest methods across clusters. Issue-based polarization interacted with cluster. Supporting the notion that endorsement of extreme protest methods might reflect motivated reasoning, among anti-reform participants (whose camp consistently protested the reform), the more extreme their views were, the more they endorsed protest methods. For pro-reform participants, the more extreme their views were, the less they endorsed protest methods.

Protest control. Issue-based and affective polarization interacted with cluster. Supporting the notion that endorsement of extreme protest control might be motivated, for anti-reform participants, the more extreme their views on the reform were, the less they endorsed protest control. For pro-reform participants, the more extreme their views were and the more negatively they felt about the anti-reform camp, the more they endorsed protest control.

**Table 3 | Predicting issue-based, affective, and perceived polarization**

| Main effects | Model 1: Issue-based polarization | | | | Model 2: Affective polarization | | | | Model 3: Perceived societal polarization | | | |
|---|---|---|---|---|---|---|---|---|---|---|---|---|
| | B | 95%CI | t | p | B | 95%CI | t | p | B | 95%CI | t | p |
| Intercept | 2.41 | [2.07,2.75] | 13.79 | <0.001 | 4.03 | [3.55,4.51] | 16.44 | <0.001 | 5.70 | [5.24,6.17] | 23.97 | <0.001 |
| Gender | −0.06 | [−0.15,0.02] | −1.55 | 0.122 | 0.11 | [0.00,0.22] | 2.01 | 0.045 | 0.04 | [−0.07,0.15] | 0.73 | 0.465 |
| Age | 0.06 | [−0.03,0.15] | 1.38 | 0.169 | 0.03 | [−0.09,0.14] | 0.46 | 0.642 | 0.09 | [−0.02,0.21] | 1.58 | 0.114 |
| Education | 0.07 | [−0.01,0.16] | 1.72 | 0.085 | 0.06 | [−0.06,0.17] | 1.01 | 0.314 | 0.03 | [−0.08,0.14] | 0.59 | 0.553 |
| SES | 0.05 | [−0.04,0.14] | 1.17 | 0.242 | 0.01 | [−0.1,0.13] | 0.24 | 0.808 | −0.01 | [−0.13,0.10] | −0.23 | 0.815 |
| Religiosity | −0.06 | [−0.15,0.04] | −1.16 | 0.245 | −0.10 | [−0.23,0.03] | −1.46 | 0.145 | **−0.15** | **[−0.28,−0.03]** | **−2.42** | **0.016** |
| Universalism/Benevolence | 0.06 | [−0.03,0.14] | 1.35 | 0.177 | **−0.17** | **[−0.28,−0.06]** | **−3.05** | **0.002** | **0.17** | **[0.06,0.28]** | **3.08** | **0.002** |
| Generalized trust | 0.03 | [−0.06,0.11] | 0.60 | 0.547 | −0.11 | [−0.23,0.01] | −1.83 | 0.068 | **−0.14** | **[−0.25,−0.03]** | **−2.42** | **0.016** |
| Civic identity | 0.05 | [−0.05,0.16] | 0.98 | 0.327 | −0.01 | [−0.16,0.13] | −0.19 | 0.850 | −0.04 | [−0.19,0.10] | −0.60 | 0.548 |
| Ethno-religious identity | 0.03 | [−0.09,0.14] | 0.46 | 0.647 | −0.11 | [−0.26,0.05] | −1.39 | 0.166 | −0.03 | [−0.18,0.12] | −0.36 | 0.718 |
| Constructive patriotism | 0.03 | [−0.07,0.13] | 0.58 | 0.562 | 0.14 | [0.00,0.27] | 1.98 | 0.048 | 0.06 | [−0.07,0.20] | 0.96 | 0.337 |
| Blind patriotism | 0.02 | [−0.10,0.13] | 0.27 | 0.790 | −0.02 | [−0.17,0.14] | −0.22 | 0.827 | −0.06 | [−0.20,0.09] | −0.73 | 0.465 |
| Trust in judiciary | −0.14 | [−0.23,−0.05] | −3.09 | 0.002 | −0.15 | [−0.30,0.01] | −1.86 | 0.063 | 0.07 | [−0.08,0.22] | 0.88 | 0.377 |
| Trust in government | 0.02 | [−0.08,0.11] | 0.32 | 0.751 | 0.15 | [−0.01,0.31] | 1.90 | 0.058 | −0.15 | [−0.31,0.00] | −1.93 | 0.054 |
| Trust in parliament | −0.02 | [−0.10,0.06] | −0.46 | 0.648 | −0.10 | [−0.25,0.05] | −1.36 | 0.176 | −0.10 | [−0.25,0.04] | −1.39 | 0.166 |
| Trust in media | 0.04 | [−0.04,0.12] | 0.87 | 0.382 | 0.07 | [−0.07,0.21] | 1.00 | 0.320 | 0.04 | [−0.09,0.18] | 0.63 | 0.528 |
| Cluster | −0.14 | [−0.37,0.08] | −1.24 | 0.217 | 0.05 | [−0.27,0.37] | 0.31 | 0.754 | −0.11 | [−0.42,0.20] | −0.73 | 0.469 |
| Wave (T1 v. T2) | −0.01 | [−0.05,0.03] | −0.46 | 0.645 | | | | | | | | |
| *Interactions with cluster* | | | | | | | | | | | | |
| *Universalism/Benevolence | −0.02 | [−0.16,0.12] | −0.26 | 0.797 | −0.07 | [−0.28,0.14] | −0.63 | 0.531 | −0.13 | [−0.34,0.09] | −1.15 | 0.251 |
| Anti-Reform | 0.01 | [−0.09,0.11] | 0.26 | 0.795 | −0.18 | [−0.33,−0.03] | −2.34 | 0.020 | 0.21 | [0.05,0.36] | 2.65 | 0.008 |
| Pro-Reform | 0.00 | [−0.10,0.09] | −0.10 | 0.919 | −0.25 | [−0.40,−0.10] | −3.25 | 0.001 | 0.08 | [−0.07,0.23] | 1.05 | 0.293 |
| *Generalized trust | 0.04 | [−0.10,0.18] | 0.52 | 0.600 | −0.05 | [−0.27,0.17] | −0.41 | 0.679 | −0.16 | [−0.39,0.07] | −1.39 | 0.164 |
| Anti-Reform | −0.02 | [−0.12,0.08] | −0.46 | 0.645 | −0.11 | [−0.26,0.05] | −1.37 | 0.171 | −0.08 | [−0.23,0.08] | −0.97 | 0.334 |
| Pro-Reform | 0.02 | [−0.09,0.12] | 0.28 | 0.777 | −0.15 | [−0.31,0.01] | −1.86 | 0.063 | −0.24 | [−0.4,−0.07] | −2.83 | 0.005 |
| *Civic identity | **−0.32** | **[−0.49,−0.14]** | **−3.47** | **0.001** | −0.22 | [−0.5,0.05] | −1.59 | 0.113 | 0.01 | [−0.27,0.29] | 0.09 | 0.924 |
| Anti-Reform | **0.19** | **[0.07,0.31]** | **3.20** | **0.001** | 0.09 | [−0.09,0.27] | 1.00 | 0.319 | −0.05 | [−0.23,0.14] | −0.50 | 0.614 |
| Pro-Reform | −0.12 | [−0.26,0.01] | −1.78 | 0.075 | −0.13 | [−0.34,0.08] | −1.22 | 0.222 | −0.03 | [−0.25,0.18] | −0.31 | 0.754 |
| *Ethno-religious identity | **0.28** | **[0.07,0.50]** | **2.55** | **0.011** | 0.03 | [−0.31,0.37] | 0.17 | 0.863 | 0.14 | [−0.2,0.48] | 0.80 | 0.424 |
| Anti-Reform | −0.01 | [−0.12,0.10] | −0.12 | 0.908 | −0.10 | [−0.27,0.07] | −1.13 | 0.260 | −0.04 | [−0.21,0.14] | −0.41 | 0.681 |
| Pro-Reform | **0.28** | **[0.09,0.47]** | **2.85** | **0.005** | −0.07 | [−0.36,0.22] | −0.46 | 0.645 | 0.10 | [−0.20,0.40] | 0.68 | 0.498 |
| *Constructive patriotism | **−0.42** | **[−0.59,−0.25]** | **−4.96** | **<0.001** | **−0.29** | **[−0.55,−0.04]** | **−2.26** | **0.025** | −0.15 | [−0.42,0.11] | −1.16 | 0.246 |
| Anti-Reform | **0.24** | **[0.11,0.36]** | **3.76** | **<0.001** | **0.27** | **[0.08,0.47]** | **2.82** | **0.005** | 0.14 | [−0.05,0.34] | 1.42 | 0.157 |
| Pro-Reform | **−0.18** | **[−0.29,−0.07]** | **−3.22** | **<0.001** | −0.02 | [−0.19,0.15] | −0.23 | 0.818 | −0.01 | [−0.19,0.16] | −0.16 | 0.876 |
| *Blind patriotism | 0.13 | [−0.06,0.32] | 1.30 | 0.193 | 0.15 | [−0.15,0.44] | 0.98 | 0.326 | 0.14 | [−0.15,0.44] | 0.95 | 0.344 |
| Anti-Reform | −0.09 | [−0.21,0.04] | −1.36 | 0.174 | −0.11 | [−0.30,0.09] | −1.07 | 0.287 | −0.13 | [−0.33,0.07] | −1.31 | 0.190 |

**Table 3 (continued) | Predicting issue-based, affective, and perceived polarization**

| Main effects | Model 1: Issue-based polarization | | | | Model 2: Affective polarization | | | | Model 3: Perceived societal polarization | | | |
|---|---|---|---|---|---|---|---|---|---|---|---|---|
| | B | 95%CI | t | p | B | 95%CI | t | p | B | 95%CI | t | p |
| Pro-Reform | 0.04 | [−0.10,0.18] | 0.53 | 0.594 | 0.04 | [−0.18,0.26] | 0.37 | 0.714 | 0.01 | [−0.21,0.24] | 0.11 | 0.916 |
| *Trust in judiciary | **−0.49** | **[−0.66,−0.33]** | **−5.84** | **<0.001** | **−0.33** | **[−0.63,−0.04]** | **−2.21** | **0.028** | −0.12 | [−0.42,0.18] | −0.76 | 0.448 |
| Anti-Reform | 0.11 | [0.00,0.21] | 1.91 | 0.056 | 0.04 | [−0.15,0.24] | 0.44 | 0.663 | 0.14 | [−0.06,0.33] | 1.35 | 0.178 |
| Pro-Reform | **−0.39** | **[−0.51,−0.26]** | **−6.09** | **<0.001** | **−0.29** | **[−0.51,−0.07]** | **−2.54** | **0.011** | 0.02 | [−0.21,0.25] | 0.16 | 0.869 |
| *Trust in government | **0.59** | **[0.42,0.77]** | **6.57** | **<0.001** | **0.60** | **[0.29,0.91]** | **3.80** | **<0.001** | 0.21 | [−0.11,0.53] | 1.30 | 0.194 |
| Anti-Reform | **−0.36** | **[−0.49,−0.23]** | **−5.34** | **<0.001** | **−0.25** | **[−0.48,−0.02]** | **−2.18** | **0.030** | −0.32 | [−0.55,−0.09] | −2.69 | 0.007 |
| Pro-Reform | **0.23** | **[0.12,0.35]** | **3.92** | **<0.001** | **0.35** | **[0.14,0.56]** | **3.25** | **0.001** | −0.11 | [−0.32,0.11] | −0.97 | 0.332 |
| *Trust in parliament | 0.02 | [−0.13,0.18] | 0.32 | 0.746 | 0.11 | [−0.17,0.39] | 0.76 | 0.449 | 0.06 | [−0.22,0.35] | 0.44 | 0.661 |
| Anti-Reform | −0.03 | [−0.13,0.08] | −0.54 | 0.592 | −0.14 | [−0.33,0.06] | −1.40 | 0.163 | −0.11 | [−0.3,0.09] | −1.10 | 0.274 |
| Pro-Reform | 0.00 | [−0.11,0.10] | −0.07 | 0.943 | −0.03 | [−0.24,0.18] | −0.25 | 0.802 | −0.04 | [−0.26,0.17] | −0.41 | 0.684 |
| *Trust in media | **−0.22** | **[−0.37,−0.07]** | **−2.81** | **0.005** | **−0.42** | **[−0.69,−0.14]** | **−2.99** | **0.003** | −0.24 | [−0.52,0.03] | −1.73 | 0.085 |
| Anti-Reform | **0.17** | **[0.07,0.26]** | **3.34** | **0.001** | **0.26** | **[0.09,0.43]** | **2.97** | **0.003** | 0.15 | [−0.03,0.32] | 1.66 | 0.097 |
| Pro-Reform | −0.05 | [−0.17,0.06] | −0.87 | 0.386 | −0.16 | [−0.37,0.06] | −1.45 | 0.147 | −0.10 | [−0.32,0.12] | −0.88 | 0.379 |
| *T1 vs. T2 | 0.07 | [−0.01,0.16] | 1.64 | 0.101 | | | | | | | | |
| Anti-Reform | −0.03 | [−0.13,0.08] | −0.54 | 0.592 | | | | | | | | |
| Pro-Reform | 0.00 | [−0.11,0.10] | −0.07 | 0.943 | | | | | | | | |
| AIC | 2649.62 | | | | 1541.64 | | | | 1559.59 | | | |
| BIC | 2801.14 | | | | 1659.08 | | | | 1677.03 | | | |
| Adjusted R² | 0.25 | | | | 0.12 | | | | 0.19 | | | |
| Sigma | 0.69 | | | | 1.36 | | | | 1.15 | | | |

*Note.* The analyses included *N* = 490 complete observations. Eight participants who had missing data in the generalized trust items were excluded from this analysis. Significant effects are bolded. For variables that significantly interacted with cluster, the interaction term and simple slopes are bolded, but corresponding significant main effects are not.

**Table 4 | Predicting downstream consequences of polarization: Protest methods, protest control, conflict management strategies, and delegitimization of political opponents**

| Main effects | Protest methods | | | | Protest control | | | | Conflict management | | | | Delegitimization | | | |
|---|---|---|---|---|---|---|---|---|---|---|---|---|---|---|---|---|
| | B | 95%CI | t | p | B | 95%CI | t | p | B | 95%CI | t | p | B | 95%CI | t | p |
| Intercept | 4.25 | [4.05,4.45] | 41.48 | <0.001 | 2.51 | [2.32,2.69] | 26.71 | <0.001 | 4.97 | [4.81,5.12] | 61.36 | <0.001 | 4.76 | [4.62,4.90] | 65.34 | <0.001 |
| Gender | **-0.15** | **[-0.29,-0.01]** | **-2.17** | **0.031** | **-0.24** | **[-0.37,-0.12]** | **-3.75** | **<0.001** | 0.08 | [-0.03,0.19] | 1.42 | 0.157 | **-0.11** | **[-0.21,-0.01]** | **-2.16** | **0.031** |
| Age | -0.04 | [-0.19,0.10] | -0.60 | 0.549 | -0.10 | [-0.23,0.03] | -1.49 | 0.138 | **0.15** | **[0.04,0.27]** | **2.63** | **0.009** | **0.12** | **[0.01,0.22]** | **2.23** | **0.026** |
| Education | 0.03 | [-0.11,0.17] | 0.41 | 0.678 | -0.07 | [-0.21,0.06] | -1.10 | 0.270 | 0.06 | [-0.06,0.17] | 0.97 | 0.335 | -0.01 | [-0.11,0.09] | -0.20 | 0.839 |
| SES | -0.07 | [-0.22,0.08] | -0.95 | 0.344 | **0.18** | **[0.04,0.31]** | **2.55** | **0.011** | 0.00 | [-0.12,0.11] | -0.07 | 0.947 | -0.01 | [-0.12,0.09] | -0.24 | 0.813 |
| Religiosity | -0.16 | [-0.31,0.00] | -1.95 | 0.052 | 0.07 | [-0.07,0.22] | 0.97 | 0.332 | 0.06 | [-0.06,0.18] | 0.95 | 0.342 | **0.12** | **[0.01,0.23]** | **2.10** | **0.036** |
| Issue-based pol. | 0.09 | [-0.06,0.23] | 1.16 | 0.245 | 0.01 | [-0.12,0.15] | 0.15 | 0.880 | **-0.17** | **[-0.29,-0.06]** | **-2.93** | **0.004** | **0.36** | **[0.25,0.46]** | **6.74** | **<0.001** |
| Affective pol. | 0.06 | [-0.08,0.21] | 0.84 | 0.403 | 0.20 | [0.06,0.33] | 2.89 | 0.004 | **-0.23** | **[-0.35,-0.12]** | **-4.00** | **<0.001** | **0.69** | **[0.59,0.80]** | **13.17** | **<0.001** |
| Perceived pol. | **0.27** | **[0.12,0.42]** | **3.55** | **<0.001** | -0.03 | [-0.17,0.10] | -0.49 | 0.627 | 0.01 | [-0.11,0.12] | 0.10 | 0.924 | 0.11 | [0.01,0.22] | 2.10 | 0.036 |
| Cluster | **-2.12** | **[-2.43,-1.81]** | **-13.30** | **<0.001** | **1.25** | **[0.96,1.54]** | **8.57** | **<0.001** | **-0.43** | **[-0.67,-0.18]** | **-3.38** | **0.001** | **0.31** | **[0.09,0.53]** | **2.72** | **0.007** |
| *Interactions with cluster* | | | | | | | | | | | | | | | | |
| *Issue-based pol. | **-1.12** | **[-1.40,-0.85]** | **-8.14** | **<0.001** | **0.70** | **[0.44,0.95]** | **5.33** | **<0.001** | 0.03 | [-0.20,0.27] | 0.26 | 0.792 | 0.18 | [-0.03,0.39] | 1.73 | 0.085 |
| Anti-Reform | **0.67** | **[0.47,0.87]** | **6.45** | **<0.001** | **-0.33** | **[-0.53,-0.14]** | **-3.41** | **0.001** | -0.20 | [-0.38,-0.03] | -2.26 | 0.024 | 0.25 | [0.09,0.4] | 3.08 | 0.002 |
| Pro-Reform | **-0.45** | **[-0.63,-0.27]** | **-4.97** | **<0.001** | **0.36** | **[0.19,0.53]** | **4.17** | **<0.001** | -0.17 | [-0.33,-0.02] | -2.17 | 0.031 | 0.43 | [0.29,0.57] | 6.14 | <0.001 |
| *Affective pol. | **-0.32** | **[-0.59,-0.05]** | **-2.35** | **0.019** | **0.41** | **[0.16,0.66]** | **3.19** | **0.002** | -0.15 | [-0.38,0.08] | -1.30 | 0.193 | 0.11 | [-0.10,0.32] | 1.05 | 0.294 |
| Anti-Reform | 0.15 | [-0.04,0.33] | 1.54 | 0.123 | 0.05 | [-0.13,0.22] | 0.51 | 0.607 | -0.16 | [-0.32,0.00] | -1.97 | 0.049 | 0.65 | [0.51,0.8] | 8.96 | <0.001 |
| Pro-Reform | -0.17 | [-0.37,0.02] | -1.78 | 0.076 | **0.46** | **[0.27,0.64]** | **4.92** | **<0.001** | -0.31 | [-0.48,-0.15] | -3.71 | <0.001 | 0.76 | [0.61,0.91] | 10.07 | <0.001 |
| *Perceived pol. | -0.20 | [-0.47,0.08] | -1.42 | 0.158 | 0.22 | [-0.04,0.48] | 1.69 | 0.091 | -0.11 | [-0.35,0.12] | -0.93 | 0.351 | **-0.31** | **[-0.52,-0.10]** | **-2.88** | **0.004** |
| Anti-Reform | 0.32 | [0.12,0.52] | 3.11 | 0.002 | -0.12 | [-0.31,0.07] | -1.23 | 0.219 | 0.07 | [-0.11,0.24] | 0.77 | 0.444 | **0.29** | **[0.13,0.44]** | **3.63** | **<0.001** |
| Pro-Reform | 0.12 | [-0.06,0.31] | 1.31 | 0.192 | 0.10 | [-0.07,0.28] | 1.16 | 0.247 | -0.04 | [-0.20,0.12] | -0.54 | 0.588 | -0.02 | [-0.16,0.12] | -0.29 | 0.773 |
| AIC | 1769.48 | | | | 1715.32 | | | | 1626.93 | | | | 1464.90 | | | |
| BIC | 1828.43 | | | | 1774.27 | | | | 1685.88 | | | | 1523.53 | | | |
| Adjusted R² | 0.50 | | | | 0.30 | | | | 0.09 | | | | 0.42 | | | |
| Sigma | 1.41 | | | | 1.33 | | | | 1.22 | | | | 1.07 | | | |
| N | 488 | | | | 488 | | | | 488 | | | | 477 | | | |

*Note. The analyses included N = 488 complete observations. Significant main effects are bolded unless the interaction term is significant. Significant simple slopes are bolded if the interaction is significant.*

Conflict management strategies. Issue-based and affective polarization were associated with less favorable views of conflict management strategies across clusters. Interactions with cluster were not significant.

Delegitimization of political opponents. Issue-based and affective polarization were associated with greater delegitimization. Perceived societal polarization interacted with cluster. For the anti-reform cluster, the more participants perceived society as polarized, the more they delegitimized their political opponents.

## Discussion

We found that Israeli Jews are polarized over the judicial reform issue, reflecting two distinct camps—pro- and anti-reform. Though we observed some shifts in views of the reform as a threat to democracy, most participants did not change their views, and the observed shift was a result of a decrease in pro-reform participants' perception of threat (i.e., they became more extreme in their views). Views of the reform transcend political orientation and party loyalties, and land instead on concepts of national identity, patriotism, and institutional trust. Even though most pro-reform participants were firmly right wing, opposition to the reform emerged across the entire political spectrum[92]. The anti- (vs. pro-) reform camp exemplified more liberal values: they expressed more universalism/benevolence values, more constructive patriotism, and had higher trust in the judiciary and media. The pro- (vs. anti-) reform camp appeared more conservative: they held a more exclusive ethno-religious identity, engaged in more blind patriotism, and had higher trust in the government. We suggest that these pre-existing differences motivated the groups' opposing views on the newly emerged reform.

In the era of global democratic backsliding, we find support for the notion that trust in different democratic institutions might motivate polarized views on newly introduced political issues, especially when these issues concern democracy. The judicial reform in Israel is the government's attempt to gain power over the judiciary. Reflecting motivated reasoning, trusting the government predicted issue-based and affective polarization (both of which are directly related to the reform) in opposing directions for the two camps. We found evidence for the importance of trust in media, which predicted greater issue-based and affective polarization among the anti-reform camp. This highlights the importance of media in democratic processes, but also suggests that its reach may be limited to people who believe that the media shares their ideology[93,94].

Adding to the literature on patriotism and polarization, we found that constructive patriotism was associated with more perceived threat from the reform, whereas blind patriotism was associated with less perceived threat, indicating that patriotism plays a role in views of anti-liberal-democratic policies. Constructive patriotism also played a role in the reform-related issue-based and affective polarization. Willingness to critique one's country, often associated with beneficial outcomes (e.g., positive outgroup attitudes)[95,96], was indeed related to less issue-based polarization among the pro-reform camp. However, constructive patriotism also had a "dark side", as it likely motivated greater issue-based and affective polarization among those who oppose the reform. It is possible that constructive patriotism specifically motivates polarization over liberal-democratic policies, so that a sense of national fallibility drives both greater emphasis on liberal-democratic principles and negative emotions towards those who threaten them. Given that patriotism was not assessed in the context of the reform (i.e., assessed only at T1, before mentioning the reform), it is unlikely that anti-reform participants' constructive patriotism merely reflects criticism of the reform or the ruling government specifically. Rather, constructive patriotism and liberal-democratic values might motivate extreme views of threats to those values and dislike of people who support such anti-liberal policies.

National identity also seemed to play a motivating role, but only for issue-based polarization. Similarly to constructive patriotism, holding the more inclusive civic identity was associated with more issue-based polarization in the anti-reform camp. This camp likely regards criticism of

governmental institutions and policies as a civic duty related to their civic identity as Israelis, thus motivating more extreme views of the reform. Ethno-religious identity, on the other hand, was associated with more extreme views in the pro-reform camp. These findings indicate a connection between more inclusive social identities and liberal-democratic attitudes, and vice versa—the connection between exclusive social identities and political preferences which lean away from liberal democracy[97,98].

So far, we discussed findings supporting the notion that pre-existing ideologically based factors motived people's views on an emerging political issue, polarizing them into two camps. Our findings further demonstrate perceptions which might be motivated by people's membership in these camps—false consensus and prioritizing different democratic features. First, our results demonstrate false consensus, a self-serving motivated bias[79]. Each camp believed that a majority of fellow citizens shared their views, suggesting that people in our sample might have been motivated to believe that their opinion is supported by most of the population. Furthermore, adding both to the literature on polarization and on false consensus, we found that those who were more polarized about an issue (i.e., held more extreme views on the issue) showed greater false consensus, suggesting that this self- and group-serving bias might be motivated by extreme views. Taken together, our findings may indicate that in the context of polarization, people who hold extreme views may use the biased belief that they belong to the majority (i.e., false consensus) to motivate and justify unwillingness to compromise, animosity, and aggression towards opponents, further deepening the divide. Future research should more directly examine the consequences of false consensus in the context of polarization. A second motivated perception was related to the core issue at hand—the importance each camp assigns to core features of democracy. We found that indeed the pro-reform camp, who are currently in power (i.e., are represented by the elected government), prioritize the majority rule principle more than the anti-reform camp, who downplay the importance of this principle. Presumably, the anti-reform camp assign the lowest importance to this principle because the reform, which they perceive as a threat to democracy, was made possible based on majority rule. The lower importance that pro-reform participants assigned to other principles is also likely to be motivated by pre-existing institutional trust (separation of power, free media) and ethno-religious identity, as well as endorsement of blind patriotism (equal human rights, minority rights). Our results echo findings from candidate choice experiments[99,100] which showed that participants with strong partisan identity and desire for ingroup advantage were willing to weaken restraints on the executive (i.e., government) in favor of their ideological agendas. The notion that patriotism, identity and ideology might underline endorsement of core democratic principles paints a concerning picture of how people form their political views.

This work extends our understanding of the consequences of polarization. Focusing on immediate, rather than long-term e.g., refs. 85,88 consequences, we find that people's endorsement of extreme protest methods and protest control were motivated by polarized views of the issue. Specifically, anti-reform participants, whose camp was protesting the reform on a weekly basis, legitimized extreme protest methods (e.g., blocking roads) and delegitimized protest control (e.g., using stun grenades against protesters) the more extreme their views against the reform were. Pro-reform participants, on the other hand, delegitimized extreme protest methods and legitimized protest control the more extreme their views supporting the reform were. In the pro-reform cluster, more extreme affective polarization was associated with legitimizing aggressive protest control means, supporting the idea that emotional reaction play a role in motivated reasoning[101,102]. For conflict management strategies and delegitimization of political opponents, we found that the more extreme people's views on the reform and negative emotions towards political opponents were the less they endorsed various conflict management strategies (e.g., willingness to compromise, dialogue) and the more they delegitimized opponents. This suggests that both issue-based and affective polarization motivate people to remain in their position, driving them away from the other side, potentially undermining conflict resolution.

## Limitations

We acknowledge several methodological limitations of the current study. First, although we recruited a representative sample in terms of age and gender, we did so only for the majority Jewish population in Israel, excluding two minority groups: Ultra-Orthodox Jews and Palestinian citizens of Israel. As such, we were unable to use our findings, especially those regarding camp sizes, as indicators of actual views of the reform in the Israeli population. This prevented us from, for example, examining which camp (pro-reform, anti-reform, or both) over-estimated their own camp's size in the population. Second, our investigation was correlational, thus limiting causal inferences. Nevertheless, we can, at least partially, rely on pre-existing differences between groups in Israel e.g., ref. 52 and on the fact that these differences certainly preceded the reform, to establish temporal precedence[103]. Similarly, the reform, and therefore any polarization over this issue, preceded the downstream consequences we focused on. Specifically, protest methods and protest control referred directly to protesting the reform, conflict management strategies referred directly to resolving this conflict, and delegitimization of political opponents centered around their understanding and motives regarding the reform. This temporal precedence, in addition to the associations that we documented, provide two of three requirements for causal inference[103]. Future work should recruit a more representative sample and add the third requirement for establishing causality—disqualification of alternative explanations.

The current research was conducted shortly after the judicial reform was introduced. Although differences in institutional trust, blind versus constructive patriotism, and ethno-religious versus civic identity between groups in general, and pro- and anti-reform camps in particular, existed before the reform, our data cannot account for the underlying causes of these differences. One likely candidate is education, and more specifically Israel's education system, which is segregated by ethnic and religious group membership[104]. Future research should examine whether this segregation leads to differences in the way people construct their national identity, criticize (vs. not) their nation, trust (vs. not) democratic institutions, and understand democracy[105–108]. Another avenue for future research is to examine whether and how segregated education systems contribute to issue-based and affective polarization.

## Conclusion

The current work shows that polarization over a specific issue can develop rapidly when based on pre-existing positions that motivate it. People's polarized views are in turn associated with self- and group-serving perceptions and with endorsement of actions that match their ideology. Motivated cognitive processes are likely playing a role in the global phenomenon of democratic backsliding by contributing to the ways people prioritize different democratic principles and therefore how they understand democracy at its core.

## Data availability

Anonymized raw data is available on OSF.

## Code availability

R-code used to conduct the current analyses in R software 4.3.0 with RStudio 2023.03.1 is available on OSF.

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

## Acknowledgements
The authors received no specific funding for this work.

## Author contributions
D.S.: Conceptualization, data curation, formal analyses, methodology, writing—original draft, writing—review & editing. A.D.: Conceptualization, data curation, formal analyses, funding acquisition, methodology, visualization, supervision, writing—original draft, writing—review & editing. M.K.: Conceptualization, data curation, formal analyses, funding acquisition, methodology, visualization, supervision, writing—original draft, writing—review & editing.

## Funding

## Competing interests
The authors declare no competing interests.
