## [Peer Review File · Communications Psychology]

17th Nov 23

Dear Dr Simunovic,

Thank you for your patience during the peer-review process. Your manuscript titled "Polarization Over the Unfolding 2023 Judicial Reform in Israel: A Real-Time Snapshot." has now been seen by 3 reviewers, and I include their comments at the end of this message. They find your work of interest, but raised some important points. We are interested in the possibility of publishing your study in *Communications Psychology*, but would like to consider your responses to these concerns and assess a revised manuscript before we make a final decision on publication.

We therefore invite you to revise and resubmit your manuscript, along with a point-by-point response to the reviewers. Please highlight all changes in the manuscript text file.

All reviewers raise points in regard to the conceptualization and framing of your work, which we hope will enable you to strengthen and clarify your narrative. The stated aims of the study should follow from the stated aims in the preregistration. Please ensure previous literature is discussed in sufficient depth (more on this below). It needs to be clear which hypotheses arose from the literature (and what literature) and which ones (if any) did not.

The reviewer comments also highlight that the Methods and Results must be more clearly presented. Please revise the section order to be aligned with the *Communications Psychology Research Article* format: Abstract - Introduction - Methods - Results - Discussion. Both Methods and Results should link tightly back to the hypotheses. Please note that for transparency, all preregistered analyses should be in the main manuscript. It should be signposted to the reader which tests were preregistered, and which were exploratory (which may include the additional analyses provided in response to the reviewer reports). We impose no word limit on Methods (a lot of the content that belongs in the Methods is currently in the Results) and are flexible about the word limit of the Results section. Only ancillary tests/results or sanity checks should be included in the Supplementary Information file rather than in the Results. No new results may be introduced in the Discussion. Any result that is interpreted in the Discussion must be in the Results section.

As Reviewer #1 mentions, some of the reported results do not reach conventional levels of statistical significance, without any evidence that you a priori set your alpha level to a value other than .05. In general, statistics reporting and interpretation needs to comply with our statistical guidelines (<https://www.nature.com/commspsychol/submit/submission-guidelines#statistical-guidelines>). In particular, marginally significant results may be stated but not interpreted. Non-significant findings derived from NHST may not be interpreted. If you wish to discuss the absence of an effect or difference, you must provide suitable statistics such as Bayesian statistics or equivalence tests in support.

Reviewer #2 raises the issue of how members of specific groups are referred to. As this pertains to self-declared group membership, we ask you to report the categories that respondents chose for themselves (in the original language, i.e. Hebrew) and provide direct English translations (<https://www.nature.com/commspsychol/editorial-policies/ethics-and-biosecurity#research-on-human-populations>).

The reviewers make many constructive suggestions as to how the Discussion may be improved. As you integrate this feedback, please keep the following general guidelines, which apply to all of our manuscripts, in mind.

We generally discourage any novelty claims and self-evaluative statements. We believe that the advance and importance of a study is best left to the reader. In your case, the reviewers' comments indicate that the importance of the work will be apparent to the reader (although its contribution may be clarified). In a similar vein, we consider it fully apparent why it is important and interesting to conduct this type of research in Israel (or other countries that face the issue of polarization). Reviewer #3 highlights that polarization is a global phenomenon and you may wish to emphasize this. As mentioned above in the context of the reviewers' request for greater clarity, existing relevant literature (which will likely consist largely of studies from the UK and the US) should be incorporated and your Discussion should explain how your study extends, confirms, or contradicts previous findings (regardless of the country of origin). As the referees highlight, it should be clear what can be learned from the study. However, it is not necessary to justify the study itself in revision by including extended speculation about generalization to other countries, or by presenting potential sources of differences between countries that were not part of the motivation to run the study a priori. As a general rule, we recommend limiting speculation in the Discussion, avoiding inferences on the suitability of possible interventions. We do mandate the inclusion of a Limitations section.

Please note that your revised manuscript must comply with our formatting and reporting requirements, which are summarized on the following checklist: Communications Psychology formatting checklist and also in our style and formatting guide Communications Psychology formatting guide.

Failure to comply with our formatting guidelines, in particular with regard to the order of the manuscript sections, the statistics reporting, and the interpretation of statistics repeatedly leads to delays in peer review, which is why the manuscript may be returned to you without review for further revisions if it is not compliant.

Please use the following link to submit your revised manuscript, point-by-point response to the referees' comments (which should be in a separate document to any cover letter) and the completed checklist:
[link redacted]

Please do not hesitate to contact me if you have any questions or would like to discuss these revisions further. We look forward to seeing the revised manuscript and thank you for the opportunity to review your work.

Best regards,

Antonia Eisenkoeck

Antonia Eisenkoeck
Senior Editor
Communications Psychology

EDITORIAL POLICIES AND FORMATTING

Editorial Policy: Policy requirements (Download the link to your computer as a PDF.)

* **CODE AVAILABILITY:** All Communications Psychology manuscripts must include a section titled "Code Availability" at the end of the methods section. In the event of publication, we require that the custom analysis code supporting your conclusions is made available in a publicly accessible repository; at publication, we ask you to choose a repository that provides a DOI for the code; the link to the repository and the DOI will need to be included in the Code Availability statement. Publication as Supplementary Information will not suffice. We ask you to prepare code at this stage, to avoid delays later on in the process.

* **DATA AVAILABILITY:**

All Communications Psychology manuscripts must include a section titled "Data Availability" at the end of the Methods section or main text (if no Methods). More information on this policy, is available at <http://www.nature.com/authors/policies/data/data-availability-statements-data-citations.pdf>.

At a minimum the Data availability statement must explain how the data can be obtained and whether there are any restrictions on data sharing. Communications Psychology strongly endorses open sharing of data. If you do make your data openly available, please include in the statement:

We recommend submitting the data to discipline-specific, community-recognized repositories, where possible and a list of recommended repositories is provided at

<http://www.nature.com/sdata/policies/repositories>.

If a community resource is unavailable, data can be submitted to generalist repositories such as figshare or Dryad Digital Repository. Please provide a unique identifier for the data (for example a DOI or a permanent URL) in the data availability statement, if possible. If the repository does not provide identifiers, we encourage authors to supply the search terms that will return the data. For data that have been obtained from publicly available sources, please provide a URL and the specific data product name in the data availability statement. Data with a DOI should be further cited in the methods reference section.

REVIEWERS' EXPERTISE:

Reviewer #1: polarization

Reviewer #2: Israeli politics & society

Reviewer #3: polarization

REVIEWERS' COMMENTS:

Reviewer #1 (Remarks to the Author):

I want to thank you for the opportunity to review this paper. The paper focuses timely and very important issue and provides insights regarding the antecedents and consequences of polarization in an urgent context. However, this study also bears some important limitations that need to be carefully addressed.

- 1) Authors should tone down on the representativeness of their sample to the general Israel population throughout the manuscript as their sample did not cover most minority groups in Israel.
- 2) I appreciate the authors' efforts to acknowledge the potential two-way relationship between trust and polarization. However, after this acknowledgement, I am not convinced of the rationale behind the decision to approach trust as a predictor and polarization as an outcome. I would also approach this relationship in the same way but still, their decision should be justified better.
- 3) Authors define and measure civic identity through being "Israeli" and ethno-religious identity through being "Jewish". Considering the historical reality and their sample, it is questionable to what extent changing names makes identities inclusive/civic or exclusive/ethnoreligious. I believe this historical/contextual reality and content(s) of national identity in Israel should be addressed in the

discussion as a potential limitation and future avenue.

4) The inclusion of the concepts that authors called “related psychological characteristics – generalized trust and universalism-benevolence values” seems arbitrary. There is no literature, introduction, or rationale for including these variables in the manuscript. These concepts and their relation with research questions should be introduced better and clearly defined in the introduction section.

5) I appreciate the authors’ effort to focus on both antecedents and consequences of polarization. I believe their results regarding to consequences of polarization are very important. As the authors mostly focus on antecedent parts in the introduction, I would highlight the consequences of polarization in the abstract and introduction. I would polish this part of the research as an importance/novelty of the study.

6) Authors should explain their rationale in terms of the order of the variables in the model. Especially why they add first patriotism and then trust?

7) In the results section, authors misleadingly discuss non-significant relationships. For instance, “For anti-reform participants, more trust in the judiciary was associated with more polarization...”. I would suggest dropping these arguments if they are not significant.

8) I believe between and within-camp polarizations should be framed better. If I understand correctly, authors first discuss the antecedents of general polarization in the population (between camps) and then conduct cluster analysis (within-camp). Especially the reason for these analyses and the summary of these findings are not clear enough as clear as the first part.

9) Authors cluster camps based on mid-points of the scale (4). I wonder why they did not choose to do it based on the mean score. Would it not be more precise clustering?

10) For me, one of the most important findings and differences between anti and pro-reform camps is that pro-reform camps prioritized majority rule over other democratic features such as protecting minorities. It should be highlighted more in the abstract, results, and discussion.

11) Although I agree with the discussion over education, I found this discussion a little bit far-fetched as it is not possible to reach this conclusion based on your design and results. I suggest toning down this argument.

Reviewer #2 (Remarks to the Author):

I really liked this MS, as it discusses an interesting and relevant topic, with some very sound analyses and conclusions. Writing is excellent, and the narrative is compelling. I have a few concerns however.

While this is a very important issue, it is also important to clarify why this is an issue beyond just the Israeli context. You do this in the discussion, but it should be made much stronger right from the start. The fact that this is (a) something taking place in many contexts and (b) a process about the understanding of democracy and what this means is super important and should be emphasized.

There is a lot going on in terms of findings, and while they are all interesting, it may help the reader to have sub-headings with the results and some narrative explanations for each domain.

Theoretically, there could be a little more background on what has been found previously (such as trust in government as well as the different types of patriotism) and how this is new.

Small issues: It is worth stating that rather than the will of the electorate, it is in fact the will of a very slight majority, with just under 50% of the population not having supported this government. Nowadays, Israeli Arabs should be referred to as Palestinian citizens of Israel.

Reviewer #3 (Remarks to the Author):

I think this is an important study that has timely implications. I enjoyed reading the introduction and felt it did a lot to inform readers about the specific context, which is important because many readers may not know much about this (and may ignore it because they are more focused on the current war). The authors have done a huge amount of work to explore and report a large number of variables, relations, and interactions. I applaud their diligence in reporting all of this within a condensed space, this is not easy!!

I have some overall conceptual feedback, and I also provide many of the comments that I made while reviewing the paper. These could be suggestions for revision or questions to be addressed.

First, while I enjoyed the narrative in the introduction for its informativeness, unfortunately it comes across very descriptive. It is more like journalism, honing in on the specific context and not maximising narrative that explains its importance at a conceptual or theoretical level. Similarly, reporting lit review about previous relations found with variables, but not so solid story-telling about how it culminates to the broad importance of this particular study. This particular journal is a new international journal that should publish issues of broad readership and impact, so I think the authors can consider how to build up the narrative to guide the broad readership on why it is important they should read and care about this, even if readers are not interested specifically in Israel—namely what is the hook? One suggestion would be to frame this in terms of a novel type of polarisation, “process polarisation”, something like a motivated reasoning type polarisation about procedural justice. This might help to build a stronger narrative around the “why”, which could not only indicate the relevance of the study but also help the reader build a meaningful mental model along the study--this point about mental model leads to my second main feedback.

Second, the authors wrote the report very clearly, so that is not an issue. Yet because of the massive amount of variables it becomes very difficult to follow (because it comes across mainly descriptive), and frankly impossible to update a mental model about the results as new results are being fed. But if the framing is built up around a broadly important “why”, then it could help the authors streamline their reporting to a more select subset of variables. Then, the rest can be in SOM for transparency. As currently written, it is just too much to digest, and then most of it gets forgotten because the reader cannot tell what’s most salient. At the end of the results section, even before reading the discussion, the reader should already be able to say “ok! Wow, now I know” but it doesn’t come across that way yet because it’s too many pieces all at once. The reader may feel a sense of gratitude that the authors are providing so much because they are “reporting all variables (e.g., for rigorous and transparency), but then it turns out that these were actually a subset of the variables and there is not much explanation for how and why the authors chose these variables rather than others. So overall, I believe the authors can take the most meaningful pieces and put together the puzzle for the reader — - build a more concise story around a few select concepts and theory — then, when authors come to the discussion they can focus on select high-impact findings rather than explaining each results again.

I also have many comments that I made along the way. Namely, feedback given in the order presented, making comments along the way. I do this so that the authors can see how a reader might react when reading in real-time, and potentially take it to consideration when revising the manuscript. I will report the more relevant ones in the order that they come in the paper.

Intro:

- you discuss issue polarisation, but this seems more like “process polarisation” or a polarisation based on motivated reasoning about procedural justice. New note, after reading the whole paper: Perhaps this terminology would help frame the issue more conceptually and help build the narrative
- is “mass polarisation” different from “polarisation”? It seems when we talk about polarisation we are thinking of a macro-level problem (even when assessing it at micro/ individual level), so what does “mass” add to this. Could be a sentence or two to explain.
- paragraph on line 62 of page 4 seems to belong later in the document as it seems like it splits the narrative. Traditionally, I would expect a “story” first, and then tell us overview of what you did later on; “why” first, “how” later.
- Line 92: sure, trust can be a pre-existing “individual differences” variable, but it seems just as likely that the announcement of the reform spiked distrust to a level that was not there before. So I am not sure I buy the argument that trust is obviously the predictor in this study or framed as an individual differences variable (i.e., it fluctuates based on communications and behaviours of organizations, entities, leaders, etc).
- Line 124: I think the terminology “split into two camps” is useful metaphor in dialogue, but it seems later that you take it literally and split the groups into camps. I don’t know if this is ideal, even if using an empirical process to justify it. Even looking at the distributions, it seems like realistically you would have three (or four)camps, one (or two) of them being moderates. I’ll come back to this later.
- Beginning line 133: You discuss three groups of outcomes. First, can these belong to overall “dimensions” that might ease reporting if they were averaged across items? Is there a factor analysis that supports this (e.g., clustering seems more appropriate here than it did in the “two camps” case)? Second, some of this seems like dehumanisation, which could be a motivated predictor for polarisation on the reform (i.e., your main variables). This leads to third point - - further down line 140 you discuss why they are downstream outcomes. I don’t agree with this completely. Possibly the first and second groups seem to align with your downstream argument, but the third does not - - it seems more likely that the fractured intergroup relations (dehumanization, deligitimization of outgroup, aggressive attitudes toward them) provide the collective psychological backdrop that affords the drift to nationalism. Indeed the authors touch on this later in the discussion in the topic of affective polarisation (e.g., Line 375). Nothing wrong with that in my opinion.. In this paper it doesn’t seem necessary to report them anything more than correlations anyway (but see my point above about framing and conceptualisation - - this might change what type of tests—predictive versus correlational versus categorical—are most appropriate). You pretty much indicate something like this yourselves in line 142, so maybe it’s just correlations.

Results:

- Line 159: crazy to get identical N for men and women after attrition from first wave!
 - Line 170: Seems like the findings of reduced perception of threat is minimised by authors. But this is interesting too .. why would it reduce in threat? Is it just a random effect, e.g., from attrition between waves?
- Also, you give a percentage when you note "most participants" but then not when you say "driven

by a minority of participants". It should be consistent otherwise it looks like a red flag.

At first the authors pre-register change scores hypotheses. But if you no longer look at change, then why are we looking at two waves? it's not clear to this point; I don't think it would be, even if one were to read the method and supplement first. It should be explained more clearly, otherwise hard to justify the decisions. For example, do you now benefit from a multilevel analysis because it's two waves, versus just using the first larger-sample wave? Even if you do, that comes at the cost of attrition / smaller N. So why choose one over the other? AKA what does it buy us and why is it the right choice? If authors stick with the current set up, the SOM might still benefit from i) reporting the results for the first W1 full sample separately, and ii) report the pre-registered hypotheses on change scores.

- Line 178: seems like the steps should be in a different order, from more concrete to more abstract/malleable. So, step 2 should be step 4, and then current step 3 and 4 should be step 2 and 3.

- Line 194: I disagree that the distribution clearly shows two camps. Actually, there seems at least one camp that is moderate. I understand that you used empirical test to justify the clusters, but isn't this even problematic for your own argument? Namely, it groups individuals who are moderate in with those who are most polarized! It makes it no better that the moderates are clustered with folks just because they are leaning on the "same side" - since degree of polarisation is part of the point itself. And then you lose that variance when you cluster them together, obscuring meaningful relations (same as you would with a median-split). If you have some theoretical reason (beyond just empirical clustering) then, I would still — a priori — wish the moderates to be separated from the extremes, namely split into theoretically-derived tertiles or maybe quartiles.

- Line 220: not sure why this is a 3-condition variable. That is, can it be explained more clearly what the "likeminded" variable is before reporting it?

Further, it seems like you have a false consensus effect here? Not only do people think more people think as they do, but they also estimate polarization too. This is one more reason you might consider splitting to tertiles or quartiles - - do the less-polarized think everybody is less polarized and the more-polarized think everybody is more polarized? That would be false consensus and interesting to report, but you cannot get at that to address the question when it's split so rough into two (highly skewed) groups

- Line 239: it's not clear why, if you do the step-wise reporting previously, why not do it here as well? Just for lack of space doesn't seem a good justification, since that just depends on which relations are reported first (which seems a bit arbitrary anyway). One option would be to report step-wise for all of them in an SOM, and then just the final-step models in the main text?

- Line 244-245: this explanation seems like "just because". I think authors should explain it more clearly.

- Line 248: By the time we get here, it becomes very exhausting because of the main issues I raised at the outset of my feedback. Relations are reported very quickly because there are so many, and then the reader has a hard time fitting each one into the whole. One solution might be to provide an elaboration of each result in its own sentence, but that won't work because there are too many of them and it would increase the length by 1/4 probably. So, it seems to me that some choices should be made how to streamline this this manuscript (especially knowing now that the report is only a subset of the measures) - - perhaps a framing around "process polarisation" can help with that.

- Line 260: these within-cluster interactions are very difficult to understand, and difficult to know how they fit into the whole. It can seem that the authors just present “everything”, but I don’t think that is the key to maximise the impact of this paper. Rather, better conceptualisation, focus on broad relevance, and streamlining will more likely do the trick.

- Line 315: I like these speculations, but they seem more fit to a specialised journal. This may be because they don’t extrapolate to broader issues/constructs/theories (i.e., beyond Israel). This connects back to my earlier comment about broad generalisability. Seems it could be revised to have broader implications if written to discuss a global problem, while examining it in a specific context and sample.

- Line 323: this example takes a larger phenomenon and notes that the past research findings replicate in this specific context. But, why, in theory, would they *not* generalise? Would rather see the reverse: something more powerful about the current findings and its importance more broadly (e.g., is there something unique and perhaps dangerous about process polarisation? This study could speak to that: gain more impact by revealing something unique in the Israel context that has implications for global issue versus findings that seem to replicate something already shown outside Israel).

- Line 335-342: This distinction between Blind and Constructive Patriotism seems to be super interesting! As in, it could have implications for understanding “far left” mindset more globally. As in my previous comments, I think this paper could step out to embed the study and findings more in theory and constructs so that the paper can have more interest and impact (e.g., this itself could be an entire manuscript).

- Line 393: Good to discuss false consensus. I wonder if this discussion can be built up re: implications beyond mere description of the findings.

- Line 397: A bit weird to report stats in the discussion (but not report stats in the results). I understand why the authors do this but it could be simply directed to SI.

Line 410: Found it a little strange to end the paper on the topic of education when the research was not focused on education. It is a speculation, and probably a correct one, but it diverts attention to something else that isn’t framed as a “directions for future research” (which the authors don’t really give much attention to more generally - - I find this to be a major issue, since really what is the impact and implication of the study?). There is no closure on the current study.

Method

- Line 428: It is not clear how and why the authors choose the measures they report in this paper. They report a large amount of variables, which may give the readers an impression that the authors report all variables. But then this is actually a select subset of measures, which raises the questions i) why so many? ii) why not the others?

- Because it is two waves but not totally clear how the waves were used, it is not clear the extent to which it is a cross-sectional survey. If it is a cross-sectional survey then there is an opportunity to select the most conceptually important (2-3) findings and either i) invite a Wave 3 for change scores

between 1-3, 2-3; or ii) replicate at a later point on an independent sample.

- Table 2: some significant results not bolded.

- Fig 3: These are nice figures but they are difficult to see. I suggest stacking the panels vertically rather than horizontally so each can be larger.

15 January, 2024

Response to Reviewers

REVIEWERS' EXPERTISE:

Reviewer #1: polarization

Reviewer #2: Israeli politics & society

Reviewer #3: polarization

REVIEWERS' COMMENTS:

Reviewer #1 (Remarks to the Author):

I want to thank you for the opportunity to review this paper. The paper focuses timely and very important issue and provides insights regarding the antecedents and consequences of polarization in an urgent context. However, this study also bears some important limitations that need to be carefully addressed.

1. Authors should tone down on the representativeness of their sample to the general Israel population throughout the manuscript as their sample did not cover most minority groups in Israel.

In the revised paper, we clarify that our sample was (at T1) recruited to be a representative sample (in terms of age and gender) of the majority Jewish population and emphasized that this sample excluded two minority groups – Ultra-Orthodox Jews and Palestinian citizens of Israel (pp. 10). In the revised manuscript, we do not treat the sample as representative of the entire Israeli population. We now also mention the exclusion of these minority groups from the sample as a limitation of the study (pp. 25).

2. I appreciate the authors' efforts to acknowledge the potential two-way relationship between trust and polarization. However, after this acknowledgement, I am not convinced of the rationale behind the decision to approach trust as a predictor and polarization as an outcome. I would also approach this relationship in the same way but still, their decision should be justified better.

Thank you for this helpful comment. We agree with reviewer that trust in institutions would not always be the predictor and polarization would be the outcome. We now clearly explain why in the specific case of the Judicial reform, trust in institutions is much more likely to be a predictor, rather than an outcome, of polarization.

We state that:

“Although polarization can decrease institutional trust, in the case of the Israeli judiciary reform, politicized institutional (dis)trust has preceded the resulting polarization, as (dis)trust existed prior to the rolling out of the reform (January 2023), at least among certain groups. Moreover, the judicial reform has been rolled out by the government to target the judiciary. This discrete event triggered polarization over this specific issue. However, peoples' pre-existing trust in the government and the judiciary preceded and determined their initial responses to the reform, and not vice versa. Importantly, although the relationship between institutional trust and polarization likely takes the shape of a negative feedback loop (i.e., the reform also impacts institutional trust, which in turn impact views on the reform), in the case of polarization over the judicial reform institutional (dis)trust started this loop to begin with.” (pp.6)

3. Authors define and measure civic identity through being “Israeli” and ethno-religious identity through being “Jewish”. Considering the historical reality and their sample, it is questionable to what extent changing names makes identities inclusive/civic or exclusive/ethnoreligious. I believe this historical/contextual reality and content(s) of national identity in Israel should be addressed in the discussion as a potential limitation and future avenue.

Thank you for the opportunity to clarify this important issue. Whereas in many countries, nationality and citizenship (i.e., civic identity) are synonyms, this is not the case in Israel. In Israel, the citizenship (i.e., Israeli) does not correspond to nationality (Handelman, 1994). Nationality is based on a combination of ethnicity and religion (e.g., Jewish, Arab/Palestinian, Druze). Therefore, the Israeli identity is, by definition, more inclusive, as it applies to all citizens of Israel, regardless of their nationality. The ethnic-religious identity is more exclusive, as it only applies to members of a specific ethnic-religious group. Nationality is indeed a meaningful construct in Israel, recognized also by the state (e.g., the population registry in Israel includes information about each Israeli citizen’s nationality).

Due to word limit constraints, we added a short explanation to the paper:

“In Israel, nationalities are significantly confounded with ethno-religious identities (e.g., Jewish for the majority group). These identities are perceived as exclusive, genetically-based and immutable, especially by religious groups, present a relevant axis of political decision-making, and are different from the inclusive civic identity (i.e., Israeli) which is common to all citizens.” (pp. 6).

We were uncertain about how to incorporate issues of national (ethno-religious and civic) identities in the Limitation sub-section, but we would be happy to do so if the Reviewer can provide some direction. Of note, we found differences between camps in endorsement of these identities, indicating that there are meaningful differences between these two identities.

4. The inclusion of the concepts that authors called “related psychological characteristics – generalized trust and universalism-benevolence values” seems arbitrary. There is no literature, introduction, or rationale for including these variables in the manuscript. These concepts and their relation with research questions should be introduced better and clearly defined in the introduction section.

Thank you for this comment, that enabled us to properly introduce these concepts. In the revised Introduction, we state:

“So far, we introduced factors that are likely to underlie motivated views on the reform and thus contribute to polarization. Additionally, we explored relevant psychological characteristics—generalized trust and universalism-benevolence values. Previous research found a mutually reinforcing negative relationship between generalized trust and polarization. Given its documented role in polarization, we included generalized trust as a person-oriented parallel to institutional trust. Universalism and benevolence are two closely related social values which denote other-regarding preferences, prosocial orientation, and a willingness to extend such behaviours across group lines. Universalism and benevolence values have been related to more liberal, left-leaning policies across national contexts. As the judicial reform undermines liberal democratic values, we explored the potential role of endorsing universalism/benevolence values in polarization over the reform.” (pp. 7)

5. I appreciate the authors’ effort to focus on both antecedents and consequences of polarization. I believe their results regarding to consequences of polarization are very important. As the authors

mostly focus on antecedent parts in the introduction, I would highlight the consequences of polarization in the abstract and introduction. I would polish this part of the research as an importance/novelty of the study.

We thank the reviewer for this vote of confidence. Following this valuable advice, we now highlight the downstream consequences in the revised Abstract and Introduction (pp. 8), hoping it indeed communicates its importance and novelty, as we cannot directly state it in the manuscript.

6. Authors should explain their rationale in terms of the order of the variables in the model. Especially why they add first patriotism and then trust?

Based on Reviewers' suggestions and to simplify the paper, we moved these ancillary analyses to the SI (Table S2 and associate comment), as this was not the focus of our investigation (and was not pre-registered). We acknowledge in the SI that the order we used is not the only logical one.

7. In the results section, authors misleadingly discuss non-significant relationships. For instance, "For anti-reform participants, more trust in the judiciary was associated with more polarization...". I would suggest dropping these arguments if they are not significant.

Thank you for pointing this issue out. We removed all mentions and discussions of non-significant and marginal results from the text.

8. I believe between and within-camp polarizations should be framed better. If I understand correctly, authors first discuss the antecedents of general polarization in the population (between camps) and then conduct cluster analysis (within-camp). Especially the reason for these analyses and the summary of these findings are not clear enough as clear as the first part.

We regret the confusion. Please allow us to better explain these analyses.

Our first analysis corresponds to the preregistered study aim reflected in Q1 – "How do trust in institutions and patriotism relate to individuals' views on the unfolding, polarizing issue of the judicial reform?". This analysis is not yet about polarization, because it does not reflect how many people in fact hold extreme views in favor or against the reform.

As the Reviewer correctly points out, the first analysis focuses on the antecedents, though instead of general polarization, the analysis predicts different views in the population. We conducted the cluster analysis to classify participants as pro- or anti-reform. We used "cluster" (pro- vs. anti-reform) to further examine characteristics of the camps (Q2). We then indeed, as the Reviewer correctly stated, examined predictors of three different types of polarization within each camp, by adding the interaction term to the models predicting issue-based, affective, and perceived polarization (Q3).

We now clearly state this in the Results regarding the model predicting view of the reform as threat: "Of note, this model examines predictors of views and does not yet address issue-based polarization." (pp. 15). We also clarify in the Methods section that "Issue-based polarization reflects the degree to which views of the reform-as-threat are polarized (i.e., tend towards the extremes and away from the neutral position)." (pp. 12). Additionally, we now present a more concise summary of the findings regarding the predictors of polarization (pp. 19-20).

9. Authors cluster camps based on mid-points of the scale (4). I wonder why they did not choose to do it based on the mean score. Would it not be more precise clustering?

We regret the confusion. To cluster the sample, we used hierarchical clustering using the *factoextra* package in *R* (p. 15). Thus, the cluster analysis was not based on the mid-point of the scale. This analysis was also not based on mean scores.

We did use the scale mid-point in the issue-based polarization measure. We now clearly explain in the revised Methods section that:

“We used the mid-point for two reasons: (1) to avoid underestimating the possibility one cluster tends more towards the extreme ends of the scale than the other, and (2) to ease interpretation of views extremity.” (pp.12).

10. For me, one of the most important findings and differences between anti and pro-reform camps is that pro-reform camps prioritized majority rule over other democratic features such as protecting minorities. It should be highlighted more in the abstract, results, and discussion.

We completely agree and thank the reviewer for this useful suggestion. We have highlighted the importance of this finding in the Abstract, Results (pp. 18), and Discussion (pp. 24).

11. Although I agree with the discussion over education, I found this discussion a little bit far-fetched as it is not possible to reach this conclusion based on your design and results. I suggest toning down this argument.

We thank the reviewer for this comment. We also think the discussion of education is important, and we agree that we cannot draw this conclusion based on our findings. We removed this conclusion from the Discussion and now mention this idea as an avenue for future research (pp. 25-26).

Reviewer #2 (Remarks to the Author):

1. I really liked this MS, as it discusses an interesting and relevant topic, with some very sound analyses and conclusions. Writing is excellent, and the narrative is compelling. I have a few concerns however.

While this is a very important issue, it is also important to clarify why this is an issue beyond just the Israeli context. You do this in the discussion, but it should be made much stronger right from the start. The fact that this is (a) something taking place in many contexts and (b) a process about the understanding of democracy and what this means is super important and should be emphasized.

We thank the Reviewer and agree that this topic is important. Yet, we were cautious about overgeneralizing beyond Israel. We agree that similar processes have been happening in other parts of the world. In the revised Introduction we now cite a greater number of publications from various national contexts (e.g., Turkey, Hungary, USA). (p. 4). We also highlight the aspect of the differential understanding of what are the important features of democracy, suggesting that this understanding and prioritizing is motivated by people's views (pp. 7-8)

2. There is a lot going on in terms of findings, and while they are all interesting, it may help the reader to have sub-headings with the results and some narrative explanations for each domain.

We thank the Reviewer for this suggestion. We added subheadings to the Results section. Additionally, we added short narrative explanations to each sub-section in the Results.

3. Theoretically, there could be a little more background on what has been found previously (such as trust in government as well as the different types of patriotism) and how this is new.

In our extensive literature review, we were unable to find empirical quantitative work specifically on trust in government and polarization. We do, however, include more relevant background on institutional trust and polarization:

“The relationship between trust in democratic institutions and polarization has been suggested to follow a negative feedback loop. Empirical findings show that institutional trust relates to lower levels of polarization. Dovetailing on the notion of motivated reasoning, it has been suggested that institutional trust can itself be politicized and polarized, contributing to polarization in ideologically motivated ways.” (pp. 5-6)

We also highlight that little is known on the links between blind and constructive patriotism and polarization and thus our study adds to this literature (pp. 7).

Small issues:

4. It is worth stating that rather than the will of the electorate, it is in fact the will of a very slight majority, with just under 50% of the population not having supported this government.

We thank the Reviewer for this important comment. Based on the information from the Central Elections Committee (retrieved from <https://votes25.bechirot.gov.il/>), we calculated that the percentage of the valid votes that were given to the Government was 48.38%, thus less than 50% of voters supported the Government. We added this information to the paper (pp. 3).

5. Nowadays, Israeli Arabs should be referred to as Palestinian citizens of Israel.

Thank you for pointing this out. We have revised the Methods section, referring to “Palestinian citizens of Israel” (pp. 10). The same change was made in the Discussion section (pp.25).

Reviewer #3 (Remarks to the Author):

1. I think this is an important study that has timely implications. I enjoyed reading the introduction and felt it did a lot to inform readers about the specific context, which is important because many readers may not know much about this (and may ignore it because they are more focused on the current war). The authors have done a huge amount of work to explore and report a large number of variables, relations, and interactions. I applaud their diligence in reporting all of this within a condensed space, this is not easy!!

Thank you!

I have some overall conceptual feedback, and I also provide many of the comments that I made while reviewing the paper. These could be suggestions for revision or questions to be addressed.

2. First, while I enjoyed the narrative in the introduction for its informativeness, unfortunately it comes across very descriptive. It is more like journalism, honing in on the specific context and not maximising narrative that explains its importance at a conceptual or theoretical level. Similarly, reporting lit review about previous relations found with variables, but not so solid story-telling about how it culminates to the broad importance of this particular study. This particular journal is a new international journal that should publish issues of broad readership and impact, so I think the authors can consider how to build up the narrative to guide the broad readership on why it is important they should read and care about this, even if readers are not interested specifically in Israel—namely what is the hook? One suggestion would be to frame this in terms of a novel type of polarisation, “process polarisation” , something like a motivated reasoning type polarisation about procedural justice. This might help to build a stronger narrative around the “why”, which could not only indicate the relevance of the study but also help the reader build a meaningful mental model along the study--this point about mental model leads to my second main feedback.

We are truly grateful to the Reviewer for suggesting the idea of motivated reasoning as the theoretical hook. We also thank the Reviewer for suggesting the term “process polarization,” though we decided to not introduce new terminology in the current paper to avoid confusion. Based on the Reviewer’s suggestions, we now frame the narrative in terms of motivated reasoning, and we believe that it immensely improved the paper. We now also discuss our findings (e.g., higher trust in government by pro-reform participants and trust in judiciary and media by anti-reform participants, prioritization of majority rule by pro-reform participants) in terms of motivated reasoning. In line with our initial theorizing, but better framed as motivated reasoning, we now refer to (1) predictors of polarization (e.g., institutional trust, patriotism) as pre-existing factors that motivate polarization, (2) false consensus and prioritizing features of democracy as motivated perceptions, and (3) downstream consequences as, at least partly, motivated by polarized views.

3. Second, the authors wrote the report very clearly, so that is not an issue. Yet because of the massive amount of variables it becomes very difficult to follow (because it comes across mainly descriptive), and frankly impossible to update a mental model about the results as new results are being fed. But if the framing is built up around a broadly important “why”, then it could help the authors streamline their reporting to a more select subset of variables. Then, the rest can be in SOM for transparency. As currently written, it is just too much to digest, and then most of it gets forgotten because the reader cannot tell what’s most salient. At the end of the results section, even before reading the discussion, the reader should already be able to say “ok! Wow, now I know” but it doesn’t come across that way yet because it’s too many pieces all at once. The reader may feel a sense of gratitude that the authors are providing so much because they are reporting all variables (e.g., for rigorous and transparency), but then it turns out that these were actually a subset of the variables and there is not much explanation for how and why the authors chose these variables rather than others. So overall, I believe the authors can take the most meaningful pieces and put together the puzzle for the reader — - build a more concise story around a few select concepts and theory —then, when authors come to the discussion they can focus on select high-impact findings rather than explaining each results again.

We thank the Reviewer for providing this valuable advice. As per the Editor’s instructions, we are unable to move main results to the SI (we did relegate the step-wise regressions to the SI, which helped to

simplify parts of the Results section; see Table S2). Additionally, we combined some of the downstream consequences based on an EFA (as per your suggestion below), which helped streamline reporting. Finally, we believe that the motivated reasoning framing that you suggested greatly improved our ability to tell a concise story, helping the reader to build a clear mental model of our findings.

I also have many comments that I made along the way. Namely, feedback given in the order presented, making comments along the way. I do this so that the authors can see how a reader might react when reading in real-time, and potentially take it to consideration when revising the manuscript. I will report the more relevant ones in the order that they come in the paper.

Intro:

4. you discuss issue polarisation, but this seems more like “process polarisation” or a polarisation based on motivated reasoning about procedural justice. New note, after reading the whole paper: Perhaps this terminology would help frame the issue more conceptually and help build the narrative.

We believe that “process polarization” is a very interesting concept. However, in this study we merely collected data regarding the specific issue (“the reform as a threat to democracy”) and therefore we opted to use existing broad terminology (i.e., issue-based polarization) that encompasses a wide range of issues, rather than suggesting that we observe a new type of polarization. We do believe that future research should look into polarization that reflects perceptions of procedural justice.

5. is “mass polarisation” different from “polarisation”? It seems when we talk about polarisation we are thinking of a macro-level problem (even when assessing it at micro/ individual level), so what does “mass” add to this. Could be a sentence or two to explain.

We apologize for the lack of clarity in our terminology. The term mass polarization refers to polarization among the voting population (vs. elite polarization occurring between political elites). We now clarify that we address mass polarization (issue-based, affective, and perceived), and term it “polarization”, for simplicity (pp. 3).

6. paragraph on line 62 of page 4 seems to belong later in the document as it seems like it splits the narrative. Traditionally, I would expect a “story” first, and then tell us overview of what you did later on; “why” first, “how” later.

Thank you for this helpful suggestion. We completely agree and we have moved this paragraph (pp. 8-9).

7. Line 92: sure, trust can be a pre-existing “individual differences” variable, but it seems just as likely that the announcement of the reform spiked distrust to a level that was not there before. So I am not sure I buy the argument that trust is obviously the predictor in this study or framed as an individual differences variable (i.e., it fluctuates based on communications and behaviours of organizations, entities, leaders, etc).

We generally agree with Reviewer’s comment – it is likely that over time, the attempts to promote the judicial reform can affect institutional trust. We now clearly explain why in the specific case of the Judicial reform trust in institutions is much more likely to be a predictor, rather than an outcome, of polarization. We now clearly explain that:

“Although polarization can decrease institutional trust, in the case of the Israeli judiciary reform, politicized institutional (dis)trust has preceded the resulting polarization, as (dis)trust existed prior to the rolling out of the reform (January 2023), at least among certain groups. Moreover, the judicial reform has been rolled out by the government to target the judiciary. This discrete event triggered polarization over this specific issue. However, peoples’ pre-existing trust in the government and the judiciary preceded and determined their initial responses to the reform, and not vice versa. Importantly, although the relationship between institutional trust and polarization likely takes the shape of a negative feedback loop (i.e., the reform also impacts institutional trust, which in turn impact views on the reform), in the case of polarization over the judicial reform institutional (dis)trust started this loop to begin with.” (pp. 5-6).

Though unfortunately we do not have a baseline measure of institutional trust (prior to the reform), our data allowed us to examine whether trust in the various democratic institutions changes from T1 to T2. We found no significant changes in trust in the judiciary, $t(497) = -0.98, p = .326$, trust in the government, $t(497) = 1.57, p = .117$, and trust in the parliament, $t(497) = -0.99, p = .321$. There was a significant but rather small change in trust in media, $t(497) = -4.77, p < .001, d = 0.18$. This lends support to our decision to treat trust as a predictor rather than an outcome.

8. Line 124: I think the terminology “split into two camps” is useful metaphor in dialogue, but it seems later that you take it literally and split the groups into camps. I don’t know if this is ideal, even if using an empirical process to justify it. Even looking at the distributions, it seems like realistically you would have three (or four)camps, one (or two) of them being moderates. I’ll come back to this later.

We agree with the Reviewer that the distribution of responses to the reform-as-threat item does not show only two camps. Yet, given that the reform is a highly politicized issue, we clustered participants based on their political orientation *and* their views on the polarizing issue. We strongly believe that this combination is crucial to understand polarization regarding this issue. There are also several statistical issues that have to be taken into account, as we elaborate in response to your comment below (#15).

9. Beginning line 133: You discuss three groups of outcomes. First, can these belong to overall “dimensions” that might ease reporting if they were averaged across items? Is there a factor analysis that supports this (e.g., clustering seems more appropriate here than it did in the “two camps” case)? Second, some of this seems like dehumanisation, which could be a motivated predictor for polarisation on the reform (i.e., your main variables). This leads to third point - - further down line 140 you discuss why they are downstream outcomes. i dont agree with this completely. Possibly the first and second groups seem to align with your downstream argument, but the third does not - - it seems more likely that the fractured intergroup relations (dehumanization, deligitimization of outgroup, aggressive attitudes toward them) provide the collective psychological backdrop that affords the drift to nationalism. Indeed the authors touch on this later in the discussion in the topic of affective polarisation (e.g., Line 375). Nothing wrong with that in my opinion.. In this paper it doesn’t seem necessary to report them anything more than correlations anyway (but see my point above about framing and conceptualisation - - this might change what type of tests—predictive versus correlational versus categorical—are most appropriate). You pretty much indicate something like this yourselves in line 142, so maybe it’s just correlations.

We agree that nine downstream consequences were too many to consider. Following the Reviewer’s suggestion, we conducted an exploratory factor analysis (reported in Table S3 of the SI) which reduced the number of consequences to four factors: support of extreme protest methods, support of extreme riot control, conflict management strategies, and delegitimization of political opponents.

Regarding the second point, we did not assess dehumanization per se. Delegitimization of political opponents referred only to the issue of the reform (e.g., “They do not understand what democracy is.”) We now clearly explain it in the Methods section:

“*Delegitimization* was assessed in the context of the judicial reform. Participants rated characteristics of people who support the reform and of people who oppose it (e.g., “They do not understand what democracy is”, “They want revenge”). We used only the items reflecting delegitimization of the opposing camp. Higher scores indicated greater delegitimization.” (pp. 13).

We agree with the third point, and we were also deliberating whether to present correlation instead of regression models. We decided in favor of regression because these analyses enabled us to examine interaction with the cluster variable. These analyses are even more important given our revised framing of motivated reasoning. Even though we pre-registered variables as downstream consequences, we acknowledge that “all the “consequences” we investigated may also act as reinforcers of polarization, thus creating a positive feedback loop” (pp. 8).

Results:

10. Line 159: crazy to get identical N for men and women after attrition from first wave!

Indeed! We double-checked.

11. Line 170: Seems like the findings of reduced perception of threat is minimised by authors. But this is interesting too .. why would it reduce in threat? Is it just a random effect, e.g., from attrition between waves?

We agree that this finding is interesting. “To further explore the observed decrease in perception of threat found in the “predicting view of the reform as threat” analysis, we conducted a repeated-measured ANOVA with a 2 (cluster: pro, anti) x 2 (time: T1, T2) on views of the reform (also see figure below) . We found a main effect of cluster $F(1, 496) = 1199.81, p < .001, \eta_p^2 = .71$, and a main effect of time, $F(1, 496) = 13.99, p < .001, \eta_p^2 = .03$, indicating that the anti-reform cluster viewed the reform as more threatening and that the overall perceived threat decreased from T1 to T2. These main effects were qualified by a cluster x time interaction, $F(1, 496) = 5.21, p = .023, \eta_p^2 = .01$. Pairwise comparisons indicated the shift was driven by pro-reform participants ($M_{T1} = 2.48, SD = 1.84; M_{T2} = 2.09, SD = 1.48$), $t = 4.15, p < .001, \eta_p^2 = .034$. There was no significant shift among anti-reform participants ($M_{T1} = 6.19, SD = 1.17; M_{T2} = 6.09, SD = 1.22$), $t = 1.06, p = .290, \eta_p^2 = .002$.” (p. 16) This analysis is reported in the Shifts in views by cluster sub section of the results.

We do not think the change is due to attrition, but of course we cannot overrule this alternative explanation. We thank the Reviewer for pointing us in this direction.

We also compared the mean views of the reform as a threat of participants who dropped from the study at T2 to the mean views of participants who remained in the study and found no significant differences between them, $t(183.69) = 1.24, p = .217$. This suggests that attrition is probably cannot account for the minor change in views we observed.

12. Also, you give a percentage when you note "most participants" but then not when you say "driven by a minority of participants". It should be consistent otherwise it looks like a red flag.

Thank you for spotting this oversight. We now added the information on this minority of participants (10.25%) who changed more drastically on their view of the reform (pp. 14).

13. At first the authors pre-register change scores hypotheses. But if you no longer look at change, then why are we looking at two waves? it's not clear to this point; I don't think it would be, even if one were to read the method and supplement first. It should be explained more clearly, otherwise hard to justify the decisions. For example, do you now benefit from a multilevel analysis because it's two waves, versus just using the first larger-sample wave? Even If you do, that comes at the cost of attrition / smaller N. So why choose one over the other? AKA what does it buy us and why is it the right choice? If authors stick with the current set up, the SOM might still benefit from i) reporting the results for the first W1 full sample separately, and ii) report the pre-registered hypotheses on change scores.

Thank you for this comment, that provides us the opportunity to clarify. First, not all variables were measured in both waves (only view of the reform as a threat and institutional trust were measured in both). Importantly, our outcome variables – affective polarization, perceived polarization, and the downstream consequences were measure in T2 and therefore conducting analyses with T1 data is impossible to answer our novel research questions.

As we measured institutional trust, blind and constructive patriotism, civic and ethno-religious identities, and adherence to universalism/benevolence values in T1, we can use them to predict views of the reform as threat with a larger sample, as the Reviewer suggested. That said, we do not think this will add much to the already lengthy set of analyses. If the Editor believes this is a valuable addition, we are happy to include it.

Second, as this study was exploratory, we pre-registered research questions/aims, but no hypotheses. Namely, one of our motivations when running T2 was to examine whether a possible change in views occurred and whether we can explain a change with our predictors. When analyzing the data, we realized that the change was very minor, and shifted the focus to our novel research questions regarding motivated perceptions and immediate downstream consequences. Because the question regarding change was preregistered, we added a more appropriate multi-level analysis (i.e., with institutional trust by wave interactions) to explain the minor change in views we did observe. We did not find any significant interactions (pp. 15). We also added the ANOVA mentioned above, showing that the minor change was driven by the pro-reform camp (pp. 16).

14. Line 178: seems like the steps should be in a different order, from more concrete to more abstract/malleable. So, step 2 should be step 4, and then current step 3 and 4 should be step 2 and 3.

Based on this and Reviewer 1's suggestions, we now only report a single model that includes all predictors (the ancillary step-wise analyses were moved to the supplementary materials; see table S2). We agree that there are other possible ways to order the predictors. As this was not the focus of our investigation (and was not pre-registered) and because this particulate order may not be the only logical one, we decided to relegate these models to the SI. Our initial reasoning was to start with demographics (i.e., common control variables) and add the more important political affiliation next. The rest of the models proceeded from person characteristics (universalism/benevolence and generalized trust) to group-level attitudes (social identity, patriotism, trust in institutions).

15. Line 194: I disagree that the distribution clearly shows two camps. Actually, there seems at least one camp that is moderate. I understand that you used empirical test to justify the clusters, but

isn't this even problematic for your own argument? Namely, it groups individuals who are moderate in with those who are most polarized! It makes it no better that the moderates are clustered with folks just because they are leaning on the "same side" since degree of polarisation is part of the point itself. And then you lose that variance when you cluster them together, obscuring meaningful relations (same as you would with a median-split). If you have some theoretical reason (beyond just empirical clustering) then, I would still — a priori — wish the moderates to be separated from the extremes, namely split into theoretically-derived tertiles or maybe quartiles.

We agree with the Reviewer that the distribution of responses to the reform-as-threat item does not show only two camps. We removed the sentence implying it (line 194 in the previous version) from the revised manuscript. Importantly, we did not cluster participants based solely on this distribution. Although it was possible that the reform created more than two camps, our data driven approach (based on participants' responses to reform-as-threat and their reported political orientation) showed that there are two distinct camps (including dendrogram and Elbow method reported in the revised SI; Figure S2).

We agree that having more (vs. less) camps will increase the variance in the camp variable, which might enable some interesting predictions. However, we are interested in the variance within each camp, which enables us to predict downstream consequences of polarization variables for each camp. For this we needed more variance within camps. Splitting the sample into several camps, even based on theoretical grounds, will drastically reduce within camp variance. As mentioned above, our data strongly suggests only two camps, therefore we did not attempt to split the sample into more camps.

16. Line 220: not sure why this is a 3-condition variable. That is, can it be explained more clearly what the "likeminded" variable is before reporting it?

Further, it seems like you have a false consensus effect here? Not only do people think more people think as they do, but they also estimate polarization too. This is one more reason you might consider splitting to tertiles or quartiles - - do the less-polarized think everybody is less polarized and the more-polarized think everybody is more polarized? That would be false consensus and interesting to report, but you cannot get at that to address the question when it's split so rough into two (highly skewed) groups

We agree that this is a false consensus effect, and we apologize for the confusing description in the previous version of the manuscript. We renamed the variable to *Estimated camp size* for clarity. In the revised Methods section, we now clearly explain how this variable was assessed:

"To assess *false consensus*, we looked at participants' perceptions of the distribution of opinions about the reform in the population. Participants indicated the percentage of the population that supports, opposes, and has no opinion about the reform (adding up to 100). False consensus manifests when participants estimate their own camp (either pro- or anti-reform) as larger than the other camps." (p. 12) The Reviewer raised a very interesting question we did not consider – "do the less-polarized think everybody is less polarized and the more-polarized think everybody is more polarized?" This led us to run a different mixed model ANOVA on estimated camp size. In the previous model, we only included cluster (anti vs. pro) as between-participants factor and estimated camp (pro, no opinion, anti) as the repeated-measures factor. In the new analysis we added issue-based polarization (0 = *neutral view*, 1 = *moderate view*, 2 = *extreme view*, 3 = *most extreme view*) as an additional between-participants factor.

First, as in the original analysis we found a significant cluster × estimated camp interaction, $F(1.98, 971.45) = 53.24, p < .001, \eta_p^2 = .10$, indicating a clear false consensus effect. Specifically, the pro-reform cluster estimated that most people support the reform ($M = 50.82\%$, $SD = 18.84$) and that about one-third oppose it ($M = 32.34\%$, $SD = 13.25$). The anti-reform cluster estimated that most people oppose

the reform ($M = 53.52\%$, $SD = 16.64$) and that about one-third support it ($M = 31.12\%$, $SD = 13.25$). Both clusters estimated the neutral opinion as equally uncommon ($M_{AntiReform} = 15.36\%$, $SD = 15.13$; $M_{ProReform} = 16.84\%$, $SD = 14.75$).

This interaction was qualified by the cluster \times issue-based polarization \times estimated camp interaction, $F(5.95, 971.45) = 10.75$, $p < .001$, $\eta_p^2 = .06$ (Figure 4). People who held extreme views expressed greater false consensus than people who were less extreme. Follow-up interaction contrasts indicated that the cluster \times estimated camp interaction, which reflects false consensus, was significant for the most extreme participants, $F(2, 490) = 138.97$, $p < .001$, and the extreme participants, $F(2, 490) = 29.21$, $p < .001$. This two-way interaction was not significant for the moderate, $F(2, 490) = 0.75$, $p = .475$, and neutral participants, $F(2, 490) = 1.93$, $p = .146$.

We think this finding is extremely interesting and we thank the Reviewer for pointing us in this direction.

Regarding the clustering of participants into pro- and anti-reform camps, it was done based on agreement with the item “the reform is a threat to democracy”, so we have no way to estimate how many people with “no opinion” on the reform in general are present in the sample. Adding the issue-based polarization to the analysis achieves what the Reviewer was aiming for, without relying on a categorization that did not emerge from the cluster analysis.

Note. Numbers in gray panels represent issue-based polarization (0 = neutral view, 1 = moderate view, 2 = extreme view, 3 = most extreme view).

17. Line 239: it’s not clear why, if you do the step-wise reporting previously, why not do it here as well? Just for lack of space doesn’t seem a good justification, since that just depends on which

relations are reported first (which seems a bit arbitrary anyway). One option would be to report step-wise for all of them in an SOM, and then just the final-step models in the main text?

We agree with this comment, and we now only report final-step models that includes all predictors for all analyses in the main text.

18. Line 244-245: this explanation seems like “just because”. I think authors should explain it more clearly.

We thank the Reviewer for the opportunity to clarify this point. In retrospect, we realized that our explanation was confusing. We now clearly explain that “To further examine differences between camps, we included interactions with cluster. This allowed us to explore the motivated associations between each predictor and polarization. For example, for the pro-reform cluster, trust in government should positively associate with issue-based polarization (i.e., view extremity), reflecting that the more they trust the government, the more extreme their view (i.e., that the reform is *not* a threat) is; for the anti-reform cluster, trust in government should negatively associate with issue-based polarization, reflecting that the more they trust the government, the less extreme their view (that the reform *is* a threat) is.” (pp. 19).

19. Line 248: By the time we get here, it becomes very exhausting because of the main issues I raised at the outset of my feedback. Relations are reported very quickly because there are so many, and then the reader has a hard time fitting each one into the whole. One solution might be to provide an elaboration of each result in its own sentence, but that won’t work because there are too many of them and it would increase the length by 1/4 probably. So, it seems to me that some choices should be made how to streamline this this manuscript (especially knowing now that the report is only a subset of the measures) - - perhaps a framing around “process polarisation” can help with that.

Following Reviewer’s previous advice, the revised Results sections is now considerably streamlined (e.g., we relegated the step-wise regressions to the SI and reduced the dimensions of the downstream consequences). We hope the revised Results section is less exhausting to read. Additionally, we followed your idea of motivated reasoning, and we believe that the current framing helps the reader to more easily follow the main findings.

20. Line 260: these within-cluster interactions are very difficult to understand, and difficult to know how they fit into the whole. It can seem that the authors just present “everything”, but I don’t think that is the key to maximise the impact of this paper. Rather, better conceptualisation, focus on broad relevance, and streamlining will more likely do the trick.

As mentioned in our response to your comment #19, we now conceptualize the paper in terms of motivated reasoning. We believe that this conceptualization allows us to streamline the findings, focus on the broad relevance of our findings, and maximizes the impact of the paper.

21. Line 315: I like these speculations, but they seem more fit to a specialised journal. This may be because they don’t extrapolate to broader issues/constructs/theories (i.e., beyond Israel). This connects back to my earlier comment about broad generalisability. Seems it could be revised to have broader implications if written to discuss a global problem, while examining it in a specific context and sample.

We agree with the Reviewer that this issue is of a broader, global importance, but it is indeed of a highly speculative nature in the context of the current research. Following the Editor’s requests, we removed all speculations from the revised Discussion.

22. Line 323: this example takes a larger phenomenon and notes that the past research findings replicate in this specific context. But, why, in theory, would they *not* generalise? Would rather see the reverse: something more powerful about the current findings and its importance more broadly (e.g., is there something unique and perhaps dangerous about process polarisation? This study could speak to that: gain more impact by revealing something unique in the Israel context that has implications for global issue versus findings that seem to replicate something already shown outside Israel).

As requested by the Editor, we removed discussion of generalization from the paper. In the revised Discussion we now focus on the main contributions of our work, including false consensus in polarization, motivated prioritizing of different democratic features, and looking at immediate downstream consequences in the context of an emerging polarizing issue.

23. Line 335-342: This distinction between Blind and Constructive Patriotism seems to be super interesting! As in, it could have implications for understanding “far left” mindset more globally. As in my previous comments, I think this paper could step out to embed the study and findings more in theory and constructs so that the paper can have more interest and impact (e.g., this itself could be an entire manuscript).

We agree that the distinction between types of patriotism is an interesting topic on its own and it can definitely be relevant for understanding “far left” mindset. Unfortunately, this interesting question is beyond the scope of our paper. We cannot address it with the data we collected, but we hope that researchers find this to be a fruitful avenue for future research.

24. Line 393: Good to discuss false consensus. I wonder if this discussion can be built up re: implications beyond mere description of the findings.

Thank you for this comment. We also think false consensus is interesting in the current context. Although we are unable to speculate about the implications of the observed false consensus effect, in the revised Discussion we elaborate on the novel findings regarding the links between false consensus and issue-based polarization:

“Our results demonstrate false consensus, a self-serving motivated bias⁷⁰. Each camp believed that their camp constitutes the majority, suggesting that people in our sample might have been motivated to believe that their opinion is supported by most of the population. Furthermore, adding to the literatures on polarization and false consensus, we found that those who were more polarized about an issue (i.e., held more extreme views on the issue) showed more false consensus, suggesting that this self-serving bias might be motivated by extreme views. Taken together, our findings may indicate that in the context of polarization, people who hold extreme views may use the biased belief that they belong to the majority (i.e., false consensus) to motivate and justify unwillingness to compromise, animosity, and aggression towards opponents, further deepening the divide. Future research should more directly examine the consequences of false consensus in the context of polarization.” (pp. 23-24).

25. Line 397: A bit weird to report stats in the discussion (but not report stats in the results). I understand why the authors do this but it could be simply directed to SI.

Thank you for pointing this out. We removed all statistics from the Discussion.

26. Line 410: Found it a little strange to end the paper on the topic of education when the research was not focused on education. It is a speculation, and probably a correct one, but it diverts

attention to something else that isn't framed as a "directions for future research" (which the authors don't really give much attention to more generally - - I find this to be a major issue, since really what is the impact and implication of the study?). There is no closure on the current study.

We now added a Limitations and Future Direction sub-section to the Discussion, where we discuss the topic of education as an important future direction for investigating polarization. We also added a conclusion at the end of the paper:

“The current work shows that polarization over a specific issue can develop rapidly when based on pre-existing positions that motivate it. People’s polarized views are in turn associated with self-serving perceptions and with endorsement of actions that match their ideology. Motivated processes are likely playing a role in the global process of democratic backsliding by contributing to the ways people prioritize different democratic principles and therefore how they understand democracy at its core.”
(pp.26)

Method

27. Line 428: It is not clear how and why the authors choose the measures they report in this paper. They report a large amount of variables, which may give the readers an impression that the authors report all variables. But then this is actually a select subset of measures, which raises the questions i) why so many? ii) why not the others?

In the spirit of open science, we reported all the variables collected in the survey as part of this pre-registration, even if they were exploratory for the purposes of other investigations. Most of these items were not relevant to the current investigation (e.g., wise reasoning).

Importantly, we included and reported all preregistered dependent variables in the paper. In retrospect, we also decided to include in our analyses perceived societal polarization as we thought it is important to capture three recognized types of polarization that are often mentioned in the literature.

28. Because it is two waves but not totally clear how the waves were used, it is not clear the extent to which it is a cross-sectional survey. If it is a cross-sectional survey then there is an opportunity to select the most conceptually important (2-3) findings and either i) invite a Wave 3 for change scores between 1-3, 2-3; or ii) replicate at a later point on an independent sample.

It is indeed a cross sectional survey regarding most of the key dependent variables as per pre-registration. Unfortunately given the current situation in Israel, a third wave as well as collecting a new independent sample are not feasible in the near future.

29. Table 2: some significant results not bolded.

We did not bold significant main effects that were qualified by a significant interaction. We are happy to do so if the Editor sees fit. We now bolded the section in the corresponding table note that explains this, for clarity.

30. Fig 3: These are nice figures but they are difficult to see. I suggest stacking the panels vertically rather than horizontally so each can be larger.

Thank you for this suggestion. We revised the presentation of the figures, now presented separately (Figures 3, 4, and 5).

7th Feb 24

Dear Dr Simunovic,

Your manuscript titled "Polarization Over the Unfolding 2023 Judicial Reform in Israel: A Real-Time Snapshot." has now been seen by our reviewers, whose comments appear below. In light of their advice I am delighted to say that we are happy, in principle, to publish a suitably revised version in *Communications Psychology* under the open access CC BY license (Creative Commons Attribution v4.0 International License).

We therefore invite you to revise your paper one last time to address the remaining concerns of our reviewers and a list of editorial requests. At the same time we ask that you edit your manuscript to comply with our format requirements and to maximise the accessibility and therefore the impact of your work.

Related to the referees' concerns, we highlight that the use of the word "camp(s)" assumes existing polarization (over the issue and in society) and in other places appears to be the interpretation of the new data; this inadvertently creates the impression of circularity in the research question. As you revise the instances raised by the reviewer, please consider revising this language as well.

SUBMISSION INFORMATION:

OPEN ACCESS:

Communications Psychology is a fully open access journal. Articles are made freely accessible on publication under a CC BY license (Creative Commons Attribution 4.0 International License). This license allows maximum dissemination and re-use of open access materials and is preferred by many research funding bodies.

For further information about article processing charges, open access funding, and advice and support from Nature Research, please visit <https://www.nature.com/commspsychol/article-processing-charges>

At acceptance, you will be provided with instructions for completing this CC BY license on behalf of all authors. This grants us the necessary permissions to publish your paper. Additionally, you will be asked to declare that all required third party permissions have been obtained, and to provide billing

information in order to pay the article-processing charge (APC).

* **DATA AVAILABILITY:**

[link redacted]

Best regards,

Antonia Eisenkoeck

Antonia Eisenkoeck
Senior Editor
Communications Psychology

REVIEWERS' COMMENTS:

Reviewer #1 (Remarks to the Author):

I am delighted to observe the significant improvements made in the updated version of the paper, which has substantially strengthened its contribution to the literature. The authors responded adeptly, addressing my concerns and suggestions with great finesse. I have no further comments to

add.

Reviewer #2 (Remarks to the Author):

I have read the revised version of this MS and believe that it is much improved and my issues have been addressed. I would still have a good proof-read to streamline some of the points, but I am satisfied that this is a significant contribution to the literature.

Reviewer #3 (Remarks to the Author):

I would like to note my appreciation for the authors' diligence in revising the manuscript and responding to reviewers' feedback. I think they have done an excellent job and the paper is much improved! Thank you to the authors.

Overall, I believe the issues have been addressed sufficiently. However, there is just a bit that could use some last touches.

Mainly, the paragraph starting page 4, line 69 (and later, paragraph starting page 5, line 98): this paragraph first says there is pre-existing polarization, and then that this reform event sparks even more polarization, however current polarization is different from the previous polarization; but then it argues that trust is what sparks the current polarization about the event, rather than the event sparking polarization (as previously noted).

So the first point is this paragraph is confusing. It seems to conflate factors or have some circular reasoning.

The second issue is that the argumentation here about trust makes too-strong claims in my view. Whether trust is pre-existing is an empirical question that is not testable in this paper; this paragraph is meant to support the use of a trust variable as antecedent "because trust is prior" but the paragraph is confusing ... it makes claims too strongly.

Then in a later paragraph beginning on page 5 line 98 the authors make the claim even more strongly, namely, "distrust started this loop to begin with". This is too strong a statement and is an empirical question that cannot be tested here.

My suggestion is, why not simply discuss trust and polarization as likely mutually reinforcing, and then discuss why it is important here to use trust as the antecedent in the context of this particular study (versus always, in any study)?

Other than that, just some double checking for typos (e.g., "pre-reform").

Thank you again to the authors for great work!